# Distinct hippocampal-cortical memory representations for experiences associated with movement versus immobility

Jai Y Yu[1,2], Kenneth Kay[1,2], Daniel F Liu[3], Irene Grossrubatscher[3], Adrianna Loback[4], Marielena Sosa[1], Jason E Chung[1], Mattias P Karlsson[1], Margaret C Larkin[1], Loren M Frank[1,2,5]*

[1]Department of Physiology, UCSF Center for Integrative Neuroscience, University of California, San Francisco, San Francisco, United States; [2]Howard Hughes Medical Institute, University of California, San Francisco, San Francisco, United States; [3]University of California, Berkeley, Berkeley, United States; [4]Princeton University, Princeton, United States; [5]Kavli Institute for Fundamental Neuroscience, University of California, San Francisco, San Francisco, United States

**Abstract** While ongoing experience proceeds continuously, memories of past experience are often recalled as episodes with defined beginnings and ends. The neural mechanisms that lead to the formation of discrete episodes from the stream of neural activity patterns representing ongoing experience are unknown. To investigate these mechanisms, we recorded neural activity in the rat hippocampus and prefrontal cortex, structures critical for memory processes. We show that during spatial navigation, hippocampal CA1 place cells maintain a continuous spatial representation across different states of motion (movement and immobility). In contrast, during sharp-wave ripples (SWRs), when representations of experience are transiently reactivated from memory, movement- and immobility-associated activity patterns are most often reactivated separately. Concurrently, distinct hippocampal reactivations of movement- or immobility-associated representations are accompanied by distinct modulation patterns in prefrontal cortex. These findings demonstrate a continuous representation of ongoing experience can be separated into independently reactivated memory representations.

DOI: https://doi.org/10.7554/eLife.27621.001

*For correspondence: loren@phy.ucsf.edu

Competing interests: The authors declare that no competing interests exist.

## Introduction

The ability to recall past experience is a fundamental cognitive function shared across species (*Crystal, 2009*; *Tulving, 1972*). Our recollections often correspond to meaningful sections or episodes taken from the continuous stream of information that was available during ongoing experience. When human subjects are asked to recall content from continuous film or written narratives with multiple themes, recall accuracy for content within each theme is independent of content from others (*Black and Bower, 1979*) and when provided with a cue, recalled information is more likely to come from the same theme than from different themes (*Boltz, 1992*; *Ezzyat and Davachi, 2011*; *Zwaan, 1996*). How ongoing and continuous representations of experience are transformed into discrete memory representations remains unclear.

The hippocampus and prefrontal cortex (PFC) maintain neural representations correlated with both ongoing experience and memory. During ongoing experience, the hippocampus encodes information about current spatial location (*Kay et al., 2016*; *O'Keefe and Nadel, 1978*; *Wilson and*

*McNaughton, 1993*) and PFC maintains information about other variables including those related to task performance (*Euston et al., 2012*; *Jung et al., 1998*; *Pratt and Mizumori, 2001*). Location specific activity in the hippocampus is interrupted intermittently by sharp-wave ripple (SWR) events, during which hippocampal activity transiently switches to expressing representations corresponding to episodes of past experience or potential future scenarios (*Buzsáki, 2015*). Hippocampal SWRs are also accompanied by coordinated changes in activity patterns in PFC (*Jadhav et al., 2016*; *Logothetis et al., 2012*; *Peyrache et al., 2009*; *Remondes and Wilson, 2015*; *Wang et al., 2015*), which reflect a coherent reinstatement of stored representations of experience. Thus, a comparison of the content of reinstated representations during SWRs with representations of ongoing experience provides an opportunity to infer how aspects of experience are maintained in memory.

To date, examination of the content of hippocampal SWR reactivation has focused exclusively on representations of experience associated with movement. These reinstated activity patterns correspond to representations of trajectories through space (*Csicsvari et al., 2007*; *Davidson et al., 2009*; *Diba and Buzsáki, 2007*; *Foster and Wilson, 2006*; *Gupta et al., 2010*; *Jackson et al., 2006*; *Karlsson and Frank, 2009*; *Pfeiffer and Foster, 2013*; *Wu and Foster, 2014*). Representation of movement trajectories that span parts or the entire environment can be reinstated through single or multiple concatenated SWRs (*Davidson et al., 2009*; *Wu and Foster, 2014*), consistent with the expression of memories for experiences that begin at one location and end at another.

Recently, hippocampal spatial coding during periods of immobility has also been identified. Location specific activity of a subset of place cells, predominantly found in CA2 but also in CA1 and CA3, persists when an animal stops. The activity of these cells is coupled to a distinct hippocampal network pattern (N-waves) not observed during movement (*Kay et al., 2016*). The distinct behavioral and neural correlates of movement- and immobility-associated experiences raise the intriguing possibility that these experiences may represent behaviorally relevant episodes that are processed differently to form distinct memory representations.

However, the reactivation of representations for immobility-associated experiences during SWRs has not yet been characterized in the hippocampus. Thus, we do not know whether immobility-associated spatial representations are reliably reactivated during the awake state. We also do not know whether such reactivations, if they exist, engage brain regions outside the hippocampus.

Since movement and immobility are fundamental features of ongoing experience, we investigated the relationship between representations of ongoing experience and memory reactivation corresponding to behavioral episodes associated with movement or immobility. We recorded activity in the CA1 region of the hippocampus and in PFC, both of which play critical roles in memory processes (*Euston et al., 2012*; *Moscovitch et al., 2016*; *Spellman et al., 2015*; *Tse et al., 2011*; *Winocur and Moscovitch, 1990*). We identified CA1 spatial representations associated with both movement and immobility and found they overlapped in time, forming a temporally continuous spatial representation of experience on the time scale of seconds. We also found immobility-associated representations are reactivated during awake SWRs. Despite the continuity of movement and immobility spatial representations, the reactivation of these representations occurred predominantly during distinct sets of SWR events. The distinction between reactivated movement- versus immobility-associated representations was also observed in the modulation of PFC activity during SWRs, consistent with a coordinated reactivation of representations associated with experience during different states of motion. Lastly, we note that the independent reactivation of movement- and immobility-associated representations could arise from a Hebbian mechanism that separates continuous representations based on the proportion of non-coincident firing between the activity of two cells (*O'Neill et al., 2008*).

## Results

### Identification of CA1 place cells participating in spatial representations during movement and immobility

We trained rats to perform a novel foraging task or a spatial alternation task (*Figure 1—figure supplement 1A*) and recorded neural activity in dorsal CA1 and PFC (*Figure 1—figure supplement 2*). We used two different tasks to examine properties of neural activity that are shared across environments with different spatial topology and task rules. In both tasks, the animals transitioned between

periods of movement (travelling between reward locations) and immobility (consuming reward) (*Figure 1—figure supplement 1B*). The animals spent similar amounts of time during movement compared with immobility (*Figure 1—figure supplement 1C*) and transitioned between these states of motion every 6–7 s (*Figure 1—figure supplement 1D*).

We first identified CA1 units participating in ongoing hippocampal spatial encoding during movement and immobility. Since immobility occurred predominantly during periods when the animals were at reward well locations (*Figure 1—figure supplement 1B*), we began by selecting units that had mean firing rates greater than 3 Hz, excluding activity during SWRs, at one or more reward well locations (*Figure 1A–C*, *Figure 1—figure supplement 3* and *Figure 1—figure supplement 4A*). This criterion was adapted from previous methods for identifying cells with place field activity (*Davidson et al., 2009*; *Frank et al., 2000*). We then asked whether their firing properties were consistent with spatial coding during immobility. We compared the properties of these cells with other CA1 units that did not fire at above 3 Hz at reward locations but that were active (>200 spikes across all sessions in a day) elsewhere in the environment (*Figure 1E–F* and *Figure 1—figure supplement 4B–E*).

The activity of CA1 units with higher firing rate during immobility at reward well locations showed strong firing specificity for single reward well locations in the tasks (*Figure 1D*). These units also spiked at times associated with lower movement speeds during the tasks (*Figure 1E*) and were active over smaller regions (*Figure 1F*) compared with CA1 units that were active elsewhere in the environment. These results demonstrate strong spatial modulation of activity in the immobility-active group of CA1 units, and we therefore refer to them as Immobility-Associated Place cells (IAPs) (n = 83) (*Figure 1A* left two columns). These IAPs have immobility spatial coding properties that are consistent with previously described N-wave coupled units, which participate in spatial coding during immobility (*Kay et al., 2016*). We relied on firing rate during immobility to identify CA1 units that participated in immobility spatial coding as we did not record, for the foraging task, neural activity in regions where N-waves are readily detected (CA2, CA3, or dentate gyrus). Although IAPs were active mostly at reward well locations, their firing characteristics are distinct from previously described goal related place cells (*Gothard et al., 1996*; *Hok et al., 2007*; *Kobayashi et al., 2003*) (see Discussion).

We refer to the remaining CA1 pyramidal units as Movement-Associated Place cells (MAPs) (n = 659) (*Figure 1A* right two columns and *Figure 1—figure supplement 4* and Materials and methods). The properties of MAPs are consistent with those of classical place cells, which have place fields located on spatial trajectories. We make the distinction between IAPs and MAPs to identify places cells that participate in spatial coding during different behavioral states and do not intend to imply any differences in cellular physiology.

## Smooth transition between CA1 movement and immobility spatial representations

Distinct hippocampal network patterns, the ~8 Hz theta rhythm and N-waves, are associated with periods of movement and immobility, respectively (*Buzsáki et al., 1983*; *Kay et al., 2016*). However, it is unknown whether transitions between movement and immobility spatial representations are smooth or abrupt. This transition could occur smoothly during transitions between movement and immobility, where the ramp down of MAP activity overlaps with the ramp up of IAP activity when animals arrive at a reward location and vice versa during departure. This would result in an overlap between the spiking of MAPs and IAPs (*Figure 2A*). Alternatively, the transition between MAP and IAP activity could be abrupt, similar to fast switches in spatial representations between consecutive theta cycles (*Jezek et al., 2011*). In this case, MAP spatial representations would be replaced by IAP spatial representations and vice versa, resulting in no overlap between MAP and IAP spiking in time (*Figure 2B*). To distinguish between these scenarios, we calculated the pairwise spike cross-correlation of MAP spiking relative to IAP spiking within a short time window (±1.5 s) centered on the time of entry to or exit from reward well locations, during which the animal transitioned between movement and immobility (*Figure 2—figure supplement 1A–B*). In the continuous spatial representation scenario, the tail of the population cross-correlation would extend beyond 0 s lag whereas in the discontinuous scenario, the tail would not (*Figure 2A–B*, second and fourth columns).

We found evidence for a smooth transition between MAP and IAP activity. MAP spiking overlapped in time with IAP activity as the animal slowed down during entry into the well location and

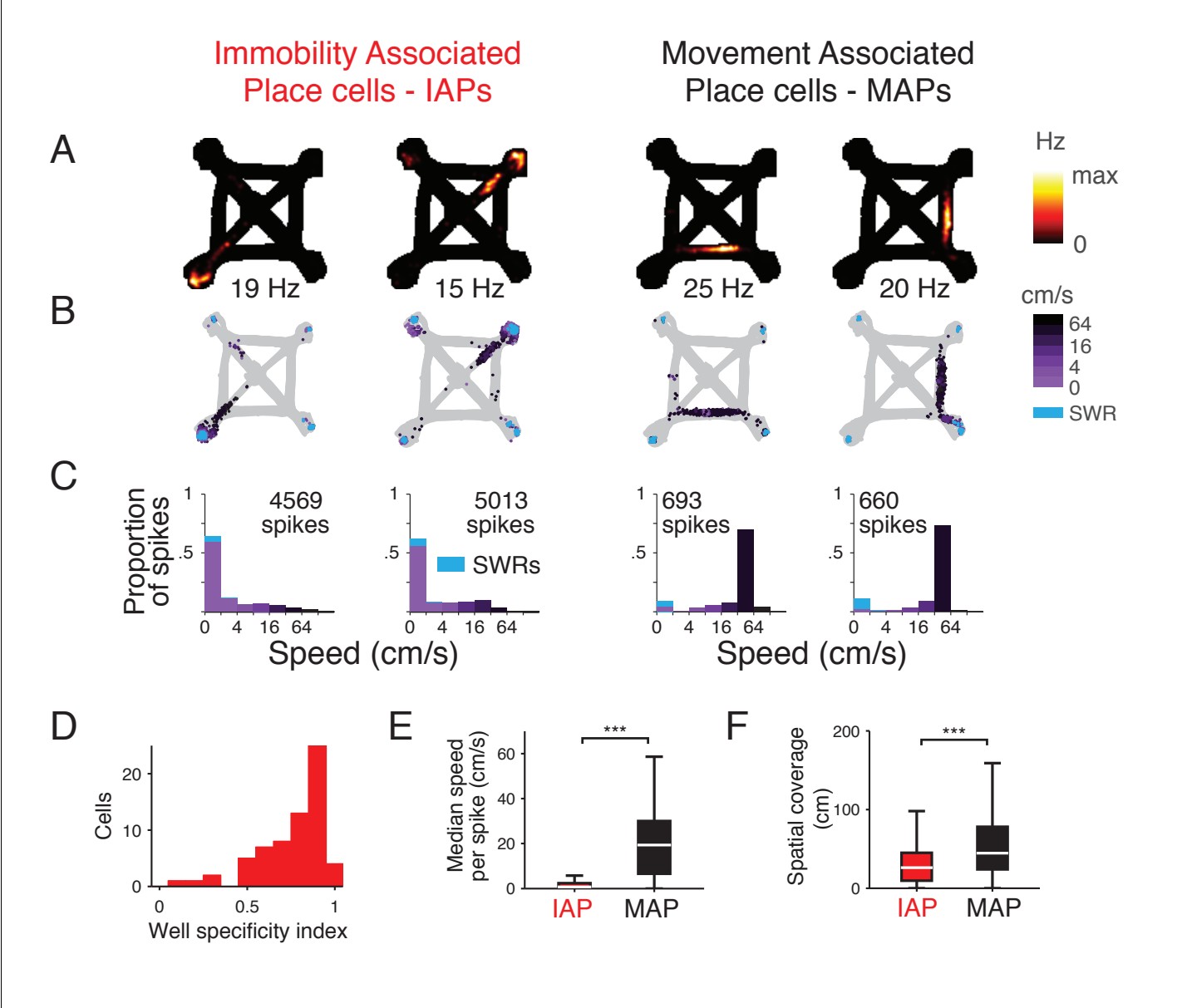

**Figure 1.** Spatial activity of CA1 place cells during movement and immobility. (A-C) Firing properties of 4 simultaneously recorded CA1 place cells from the foraging task. Two example Immobility-Associated Place cells (IAPs) are shown in the first two columns and two example Movement-Associated Place cells (MAPs) are shown in the last two columns. (A) Occupancy normalized firing rate. Maximum rate is shown below each panel. (B) Location and speed of the animal at the time of each spike. Speed is indicated by shades of purple. Spikes during SWRs, which occur at low speeds, are shown in cyan. (C) Histogram of speed for all spikes. The histogram is stacked and the proportion of spikes during SWRs is shown in cyan. Amount of time spent in each speed category is shown in *Figure 1—figure supplement 3*. (D) Distribution of well specificity indices for all IAPs. (E) Median movement speed at the times of spikes for IAPs and MAPs. Even though we selected IAPs based on mean firing rate at locations of immobility, IAPs tend to only spike at low speeds rather than being generally active at all times. Wilcoxon rank-sum test: ***$p < 10^{-4}$. (F) Spatial coverage of IAPs and MAPs. Wilcoxon rank-sum test: ***$p < 10^{-4}$.

DOI: https://doi.org/10.7554/eLife.27621.002

The following figure supplements are available for figure 1:

**Figure supplement 1.** Two spatial navigation tasks with alternating periods of movement and immobility.

DOI: https://doi.org/10.7554/eLife.27621.003

**Figure supplement 2.** Histology of subjects that performed the foraging task.

DOI: https://doi.org/10.7554/eLife.27621.004

**Figure supplement 3.** Proportion of time the animal spent in each speed category.

*Figure 1 continued on next page*

*Figure 1 continued*

DOI: https://doi.org/10.7554/eLife.27621.005

**Figure supplement 4.** Characterization of IAPs and MAPs in each task.

DOI: https://doi.org/10.7554/eLife.27621.006

similarly overlapped as the animal exited the well location (median cross-correlation shown in *Figure 2C*). These results are consistent with MAPs and IAPs participating in a continuous spatial representation on the time scale of seconds.

Interestingly, there was also a sharp dip in the average cross-correlations at 0 s lag (*Figure 2C*) with a width equal to the duration of a theta cycle, which prompted us to examine spiking on the time scale of theta oscillations (~8 Hz). We first calculated spike autocorrelations for IAPs and MAPs during the movement and immobility transition periods around the time of reward well location entry and exit. For IAP and MAP spikes that occurred when the animal was moving (speed >4 cm/s),

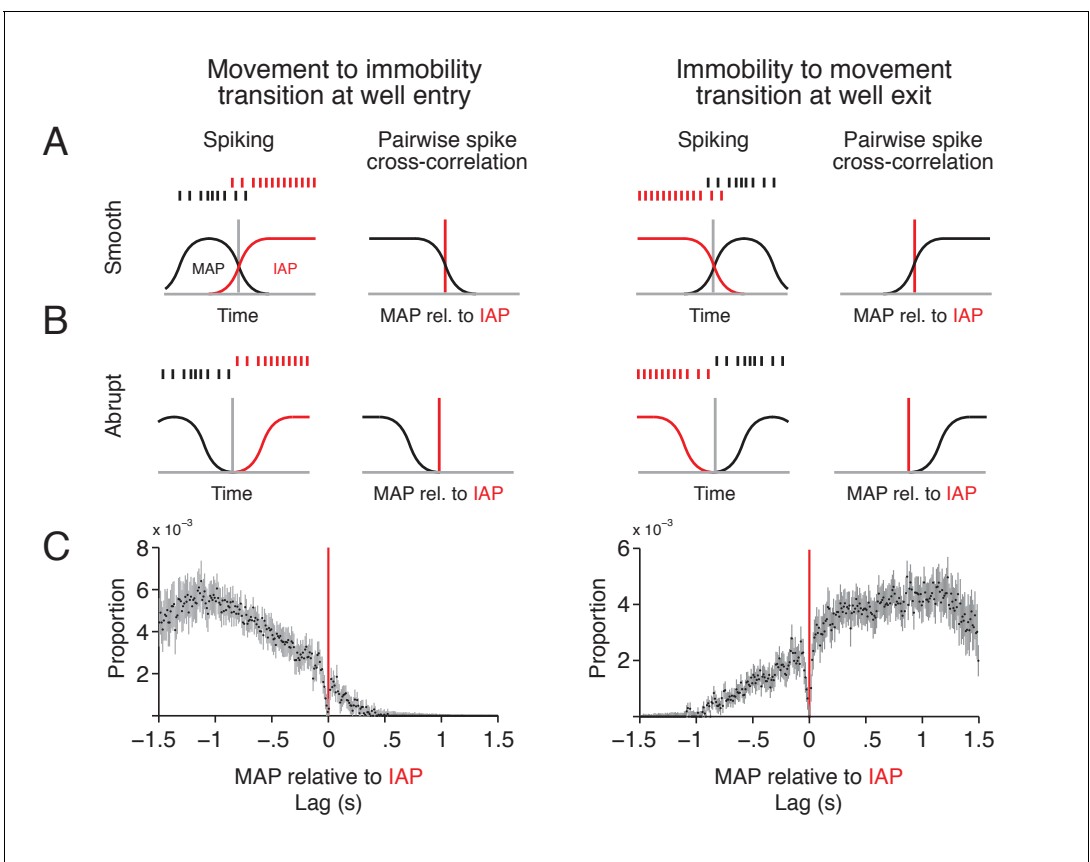

**Figure 2.** MAPs and IAPs form a continuous spatial representation during transitions between movement and immobility. (**A–B**) Schematic of two scenarios for activity of MAPs and IAPs during transitions between movement and immobility as an animal enters (left column) or exits (right column) a reward well location. For a smooth transition, where spikes overlap in time between IAPs and MAPs (**A**), the cross-correlogram should have a tail that extends beyond 0 s lag. For an abrupt transition (**B**), where spikes do not overlap in time, the tail of the distribution should end at or before 0 s lag. (**C**) Pairwise spike cross-correlogram for spiking of MAPs relative to spiking of IAPs for spikes in a ± 1.5 s window relative to reward well location entry (left, MAP-IAP pairs n = 207) and exit (right, MAP-IAP pairs n = 217). Cross-correlation is histogrammed in 10 ms bins. Data shown as median ±1.58 × IQR divided by the square root of n, which corresponds to the notch on a box plot.

DOI: https://doi.org/10.7554/eLife.27621.007

The following figure supplements are available for figure 2:

**Figure supplement 1.** Spiking of MAP and IAP can be theta modulated during transitions between movement and immobility.

DOI: https://doi.org/10.7554/eLife.27621.008

**Figure supplement 2.** Lower proportion of coincident spikes between IAP and MAP pairs compared with MAP pairs or IAP pairs.

DOI: https://doi.org/10.7554/eLife.27621.009

we found peaks in the autocorrelation consistent with entrainment to theta oscillations (*Figure 2—figure supplement 1C* left panel and *1D* right panel). The peaks were less prominent in the autocorrelation of spikes from periods when the animal was immobile (speed <1 cm/s) (*Figure 2—figure supplement 1C* right panel and *1D* left panel).

Previous work has demonstrated that MAPs with overlapping place fields spike sequentially within each theta cycle as an animal travels across their place fields. As a result, there are consistent spike timing offsets on the time scale of theta oscillations that reflect the spatial separation between a cell pair's place fields (*Dragoi and Buzsáki, 2006*; *Foster and Wilson, 2007*; *Gupta et al., 2012*; *Skaggs et al., 1996*). Our results indicate that this same pattern of coactivity is present for MAP-IAP pairs, where MAPs and IAPs can be active within individual theta cycles during transitions between immobility and movement. On the finer timescale of a theta cycle, most MAP-IAP pairs had offsets in peak cross-correlation of up to half a theta cycle (*Figure 2—figure supplement 1E–F*), which was correlated with the separation of spiking activity on a longer time scale (*Figure 2—figure supplement 1G–J*). These findings provide further evidence that movement and immobility spatial representations transition smoothly from one to the other.

This pattern of coactivity during ongoing experience seems likely to promote the strengthening of associations between MAPs and IAPs with temporally overlapping activity. At the same time, we also note that that IAP activity continued for many seconds during immobility, a period when MAPs were largely silent. We quantified this by computing, for MAP-MAP, IAP-IAP, and MAP-IAP pairs, the proportion of spikes from one unit that occurred in a ± 50 ms window of spikes from the other relative to spikes that occurred in a ± 3.3 s window (6.6 s is the duration of a typical bout of movement and immobility). This analysis revealed a lower proportion of temporally nearby spikes for MAP-IAP pairs as compared to the other pair types (*Figure 2—figure supplement 2*), and thus revealed a pattern of spiking that could lead to a weakening of associations between MAPs and IAPs.

## Separate reactivation of immobility-associated spatial representations

Does the coactivity between MAPs and IAPs during ongoing experience translate into coactivity during SWRs? Or does the larger proportion of non-adjacent spikes translate into separate SWR reactivation of movement and immobility representations? Before addressing these questions, we first noted that the reactivation of immobility-associated spatial representations during awake SWRs has not been established. Therefore, we asked whether IAP spatial representations are reactivated during awake SWRs (Materials and methods, *Figure 3—figure supplement 1*). We found that IAPs were active during a subset of awake SWRs detected during periods of immobility at reward well locations (*Figure 3A*). These time-locked increases in spiking during SWRs occurred both at the location where the IAP had its place field (in field, *Figure 3—figure supplement 2*) and during SWRs at other locations (not in field) (*Figure 3B–C*).

These reactivations also reflected the coherent reinstatement of immobility associated spatial representations. We calculated the baseline-adjusted coactivation for different cell pairs (*Cheng and Frank, 2008*), defined as the observed probability that both cells in the pair participated in the same SWR normalized by the expected probability of joint participation under an assumption of independence (see Materials and methods). We found that IAP pairs with place fields located at the same reward well location were strongly coactivated during SWRs, whereas those cell pairs with place fields located at different reward well locations showed no more coactivation than expected by chance (*Figure 3—figure supplement 3*). This indicates that representations of locations associated with immobility are reactivated individually rather than simultaneously. Further, for IAP pairs with place fields at the same well location, there was strong SWR coactivity during events that occurred away from the cells' place fields (*Figure 3—figure supplement 3*), indicating that non-local immobility spatial representations are reactivated at the ensemble level during SWRs.

We then asked whether movement and immobility spatial representations are reactivated together or individually. Strikingly, there was a clear separation between the reactivation of MAPs and IAPs. We first counted the number of SWRs that contained only MAPs (MAP-SWRs), only IAPs (IAP-SWRs) or a combination of both (MAP and IAP-SWRs) (*Figure 4A*). The large majority of SWRs contained either only MAPs (n = 12881, 67%) or only IAPs (n = 5009, 26%). Only a small fraction of events contained both MAPs and IAPs (n = 1311, 7%). Based on the probability of either MAPs or IAPs participating individually in a SWR, the observed number of MAP and IAP SWRs was

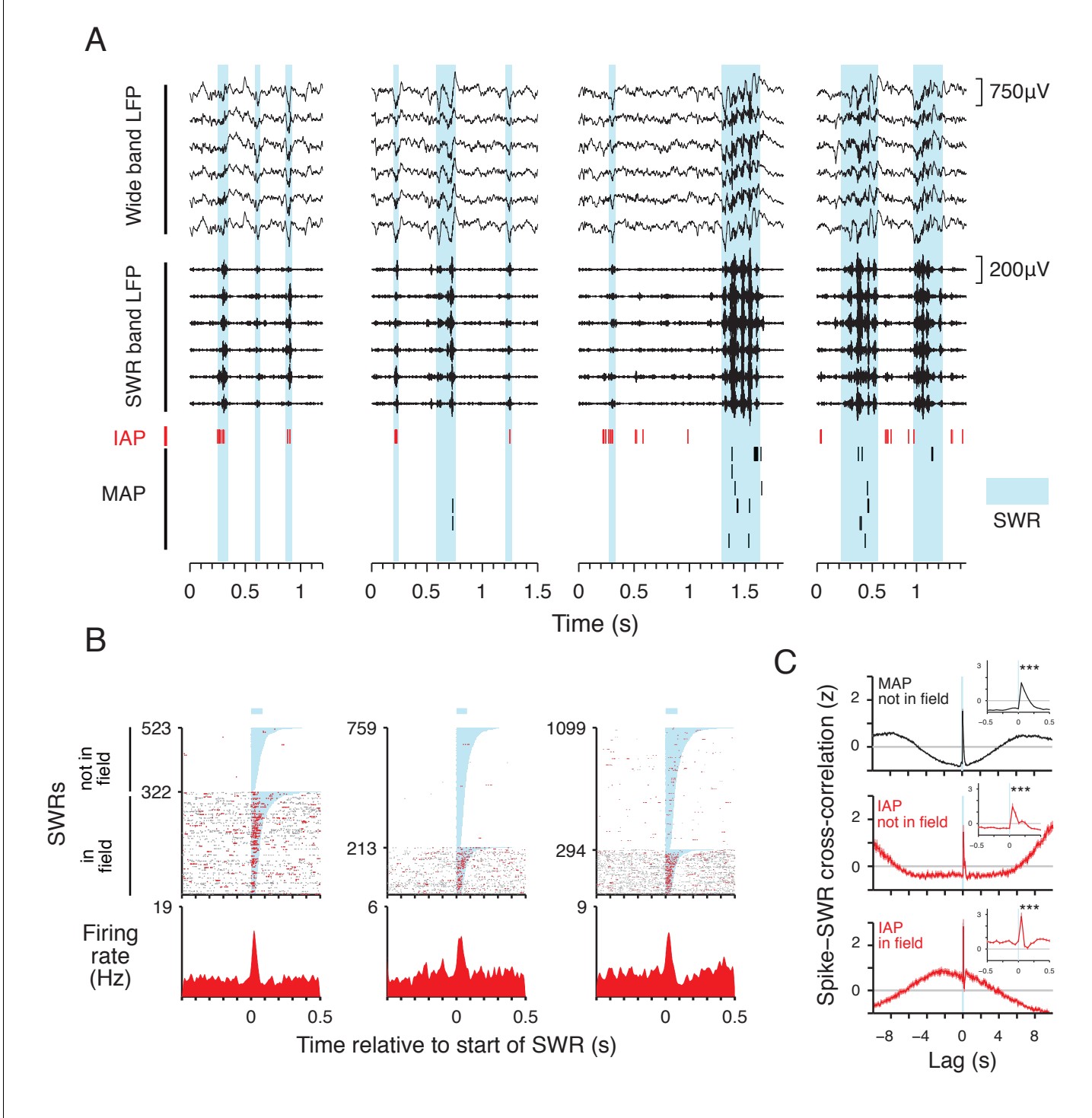

**Figure 3.** IAPs are reactivated during SWRs. (**A**) Example local field potential (LFP) and place cell spiking during SWRs (cyan). Wide band LFP (1–400 Hz) and SWR band LFP (150–250 Hz) traces from 6 simultaneously recorded sites in CA1 are shown. IAP spikes are shown in red and MAP spikes are shown in black. The four examples correspond to immobile periods at wells in the foraging task. All examples are from the same day. (**B**) Spiking activity of three IAPs aligned to the start of SWRs. Spikes within the duration of SWRs are colored red and spikes outside of SWRs are colored grey. In field events are SWRs occurring at the reward well where the IAP has a place field. Not in field events occur at other reward wells. The orders of in field and not in field SWRs are sorted separately by SWR duration (shaded cyan, mean duration indicated by bar above spike raster). Bottom: firing rate histogram shows mean firing rate across all SWRs. Leftmost example is the IAP shown in (**A**). (**C**) Normalized cross-correlation (mean ±SEM, 50 ms bins) of MAP (black, n = 589) and IAP (red, n = 77) spiking relative to start of SWRs (cyan line). For MAPs, SWRs that do not occur within the cells' place fields and are

*Figure 3 continued on next page*

*Figure 3 continued*

referred to as not in field events. For IAPs, the cross-correlation of spikes and not in field SWRs is shown in middle panel, and the cross-correlation of spikes and in field SWRs is shown in lower panel. Lag refers to time relative to the start of SWR events. Inset shows the cross-correlation in a 1 s window centered at 0 s lag. For in and not in field events, both IAPs and MAPs show significant increases in activity aligned to the start of SWRs (250 ms pre vs. 250 ms post SWR start, Wilcoxon rank-sum test: \*\*\*$p<10^{-4}$). We note that SWRs tend to occur during immobility, thus IAP activity is higher within seconds around the time of in field SWRs compared with MAPs and not in field SWRs.

DOI: https://doi.org/10.7554/eLife.27621.010

The following figure supplements are available for figure 3:

**Figure supplement 1.** SWR detection threshold determination.

DOI: https://doi.org/10.7554/eLife.27621.011

**Figure supplement 2.** Baseline spiking-SWR cross-correlation for SWRs that occurred within the place field of IAPs.

DOI: https://doi.org/10.7554/eLife.27621.012

**Figure supplement 3.** SWR coactivity for IAPs.

DOI: https://doi.org/10.7554/eLife.27621.013

significantly less than expected (expected n = 4671, 24%, actual vs. expected p<0.0001, Chi squared test, d.f. = 1). This indicates MAP and IAPs are significantly less likely to participate in the same reactivation event than expected by chance.

A pairwise analysis confirmed the largely independent reactivation of MAPs and IAPs during SWRs. Previous examinations of coactivity for MAPs have demonstrated that whether a pair of place cells participates in a SWR is related to the amount of coactivity in time outside of SWRs (*O'Neill et al., 2008*; *Singer and Frank, 2009*). To control for the effect of temporal separation in activity during the task on coactivity during SWRs, we selected pairs of place cells that were coactive within a behaviorally relevant duration, which corresponds to a pair of cells having a cross-correlation peak within 3.3 s (half the duration of a typical bout of movement and immobility, *Figure 1—figure supplement 1D* column 3). This selection ensures we are only comparing place cells that were active within similar intervals of time during the task. The coactivity distribution shows MAP-IAP pairs had significantly lower levels of SWR coactivity than MAP-MAP or IAP-IAP pairs (*Figure 4B* and *Figure 4—figure supplements 1* and *2*). We also replicated this result by using a more stringent condition, where we included only cell pairs with overlapping spikes during the transition period between movement and immobility at reward location entry or exit (*Figure 4—figure supplement 3*). These results show that given comparable activity over similar intervals of time during the task, MAPs and IAPs are strongly biased to participate in different SWRs.

## Distinct properties for SWRs involving movement and immobility representations

For spatial trajectory reactivations during SWRs, there is a positive correlation between the duration of a SWR and the spatial extent of the reactivated trajectory (*Davidson et al., 2009*; *Wu and Foster, 2014*). The activity of IAPs represented single locations during ongoing experience, which suggests that IAP-SWRs, corresponding to reactivation of immobility representations, could be shorter than those containing MAPs. We examined the properties of SWRs that contained only IAPs (IAP-SWRs), only MAPs (MAP-SWRs) or both IAPs and MAPs (MAP & IAP-SWRs) and found that IAP-SWRs were significantly shorter in duration and lower in amplitude than SWRs containing MAPs (*Figure 4C–D* and *Figure 4—figure supplement 4*), even when we controlled for the number of cells and spikes detected during each SWR (*Figure 4—figure supplement 5*, *6* and *7*). Nonetheless, IAP-SWR events were detected across multiple sites in CA1 (*Figure 3A*, see Materials and methods) and contained the characteristic increase in local field potential (LFP) power in the gamma frequency (~30 Hz) seen during SWRs (*Carr et al., 2012*) (*Figure 4D*), supporting their identification as SWRs.

We did not observe any strong relationships in the relative timing between IAP-SWRs and MAP-SWRs, indicating these events were temporally interspersed (*Figure 4—figure supplement 8*). We did notice MAP & IAP-SWRs were longer and higher in amplitude than the other types of SWRs (*Figure 4C–D* and *Figure 4—figure supplement 4*), and that these events may correspond to trajectory reactivations. Trajectory replay events are biased to start from the animal's current location (*Davidson et al., 2009*; *Karlsson and Frank, 2009*), which is often where the animal is immobile. This suggests that in the small minority of cases where both MAPs and IAPs were active in the same

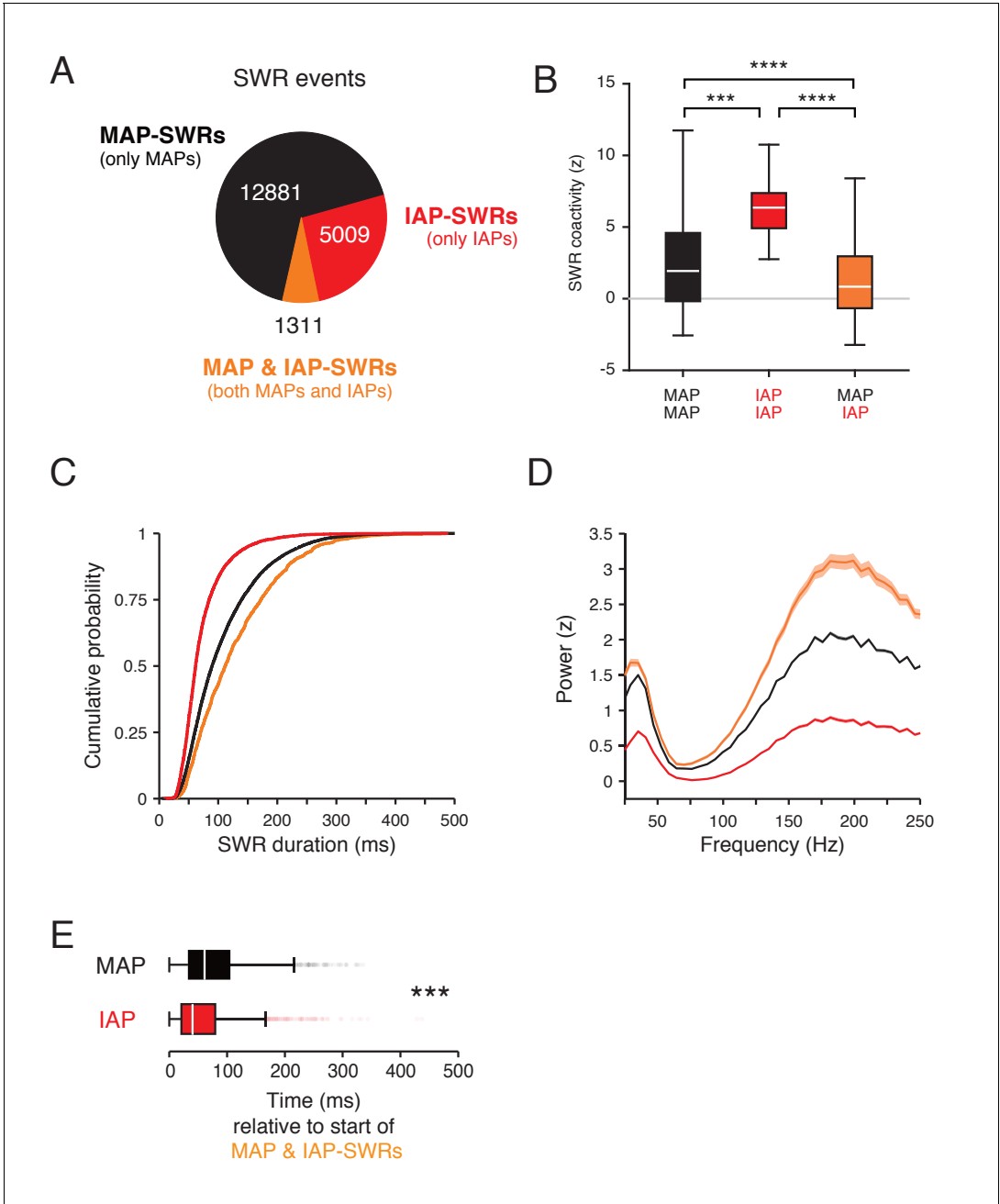

**Figure 4.** Separate SWR reactivation of IAPs and MAPs. (**A**) Count of SWRs containing MAPs and IAPs. (**B**) Coactivity during SWRs for MAP-MAP pairs (black, n = 1015) and IAP-IAP pairs (red, n = 13) is higher than IAP-MAP pairs (orange, n = 138). Wilcoxon rank-sum test: \*\*\*p<10$^{-3}$ and \*\*\*\*p<10$^{-4}$. Data from both tasks are combined. Only cell pairs that are active within half the duration of a typical trial (peak cross-correlation <3.3 s) were included. This time is derived from the time course of bouts of movement and immobility in the task (*Figure 1—figure supplement 1D*). Outliers are not shown. (**C**) Cumulative distribution of durations for SWR events in (**A**). Kolmogorov-Smirnov test: p<10$^{-4}$ for all pairwise comparisons between groups. (**D**) LFP spectral power (mean and 95% confidence of the mean) in a 100 ms window aligned to the start of SWRs normalized to all times in a task session. The mean of spectral power between 150–250 Hz is significantly different between groups. Wilcoxon rank-sum test: all pairwise comparisons p<10$^{-4}$. (**E**) Time of MAP (black) and IAP (red) spiking relative to the start of MAP and IAP-SWRs. IAPs spike earlier than MAPs during MAP and IAP-SWRs. Wilcoxon rank-sum test: \*\*\*p<10$^{-4}$.

DOI: https://doi.org/10.7554/eLife.27621.014

The following figure supplements are available for figure 4:

**Figure supplement 1.** Control for the effect of temporal proximity of spiking activity on SWR coactivity.

DOI: https://doi.org/10.7554/eLife.27621.015

*Figure 4 continued on next page*

*Figure 4 continued*

**Figure supplement 2.** Control for the effects of spatial proximity of spiking activity on SWR coactivity.
DOI: https://doi.org/10.7554/eLife.27621.016
**Figure supplement 3.** MAP and IAP pairs with temporal spiking overlap during movement and immobility transitions show lower SWR coactivity than MAP-MAP and IAP-IAP pairs.
DOI: https://doi.org/10.7554/eLife.27621.017
**Figure supplement 4.** Spectrotemporal properties of SWR groups are consistent between the foraging and the W-shaped alternation task.
DOI: https://doi.org/10.7554/eLife.27621.018
**Figure supplement 5.** Cell and spike count during SWRs.
DOI: https://doi.org/10.7554/eLife.27621.019
**Figure supplement 6.** Subsample to match cells active per SWR.
DOI: https://doi.org/10.7554/eLife.27621.020
**Figure supplement 7.** Subsample to match spikes per SWR.
DOI: https://doi.org/10.7554/eLife.27621.021
**Figure supplement 8.** Cross-correlation between MAP- and IAP-SWRs.
DOI: https://doi.org/10.7554/eLife.27621.022
**Figure supplement 9.** IAPs spike earlier than MAPs during MAP & IAP-SWRs after matching SWR durations.
DOI: https://doi.org/10.7554/eLife.27621.023

SWR, the IAPs would tend to lead. We therefore analyzed when MAPs and IAPs spiked relative to the start of MAP & IAP-SWRs. We found IAPs tended to spike earlier in the SWR than MAPs (*Figure 4E*), even after accounting for differences in SWR duration from which IAP and MAP spikes were selected from (*Figure 4—figure supplement 9*). These results are consistent with MAP and IAP-SWRs reactivating representations that begin with locations of immobility encoded by IAPs and continue with locations encoded by MAPs.

## Prefrontal cortical activity changes during SWRs distinguish between reactivations of movement and immobility representations

Is the distinction between movement- or immobility-associated reactivations restricted to the hippocampus or is the distinction observed in other regions? Hippocampal SWRs are associated with widespread neocortical activity changes (*Logothetis et al., 2012*). In PFC, changes in activity at the time of awake and sleep SWRs reflect a concurrent reinstatement of activity patterns that were present during ongoing experience (*Peyrache et al., 2009*; *Remondes and Wilson, 2015*; *Wang and Ikemoto, 2016*) and the content of activity during SWRs is coordinated between PFC and the hippocampus (*Jadhav et al., 2016*). However, analysis of the content of coordinated activity changes has so far concentrated on representations for movement-associated experiences (*Jadhav et al., 2016*) or included all SWRs (*Remondes and Wilson, 2015*). Thus, it is unclear whether there are specific cortical activity patterns that accompany the reactivation of immobility-associated hippocampal reactivations.

Having the ability to classify SWRs into MAP- or IAP-SWRs, we examined whether the distinct patterns of PFC activity changes accompany the reactivation of movement and immobility representations in the hippocampus. We analyzed data from animals performing the foraging task, where we simultaneously recorded neurons from the hippocampus and PFC. From an initial set of 712 PFC units, we selected units that were actively engaged in the task (mean firing rate >1 Hz during either periods of immobility or movement or both) and were recorded during behavioral sessions where we could identify both MAP- and IAP-SWRs. We then examined the activity of these units (n = 112) during MAP- and IAP-SWRs, which comprise the majority of SWR events.

We found ~63% of these PFC units (70/112) showed significant modulation during one or both types of SWRs. This percentage is almost twice the amount we reported previously without knowledge of different SWR types (*Jadhav et al., 2016*), which suggests that IAP-SWRs engage a set of SWR-modulated PFC units that were not observed before. Importantly, the patterns of modulation were qualitatively different depending on the type of SWR. Many PFC units showed excitation during one type of SWR and either no change in activity or inhibition during the other (*Figure 5A–B*), which was unexpected and had not been previously demonstrated. Others showed inhibition during both types of SWRs (*Figure 5C*), consistent with previous findings (*Jadhav et al., 2016*). To quantify

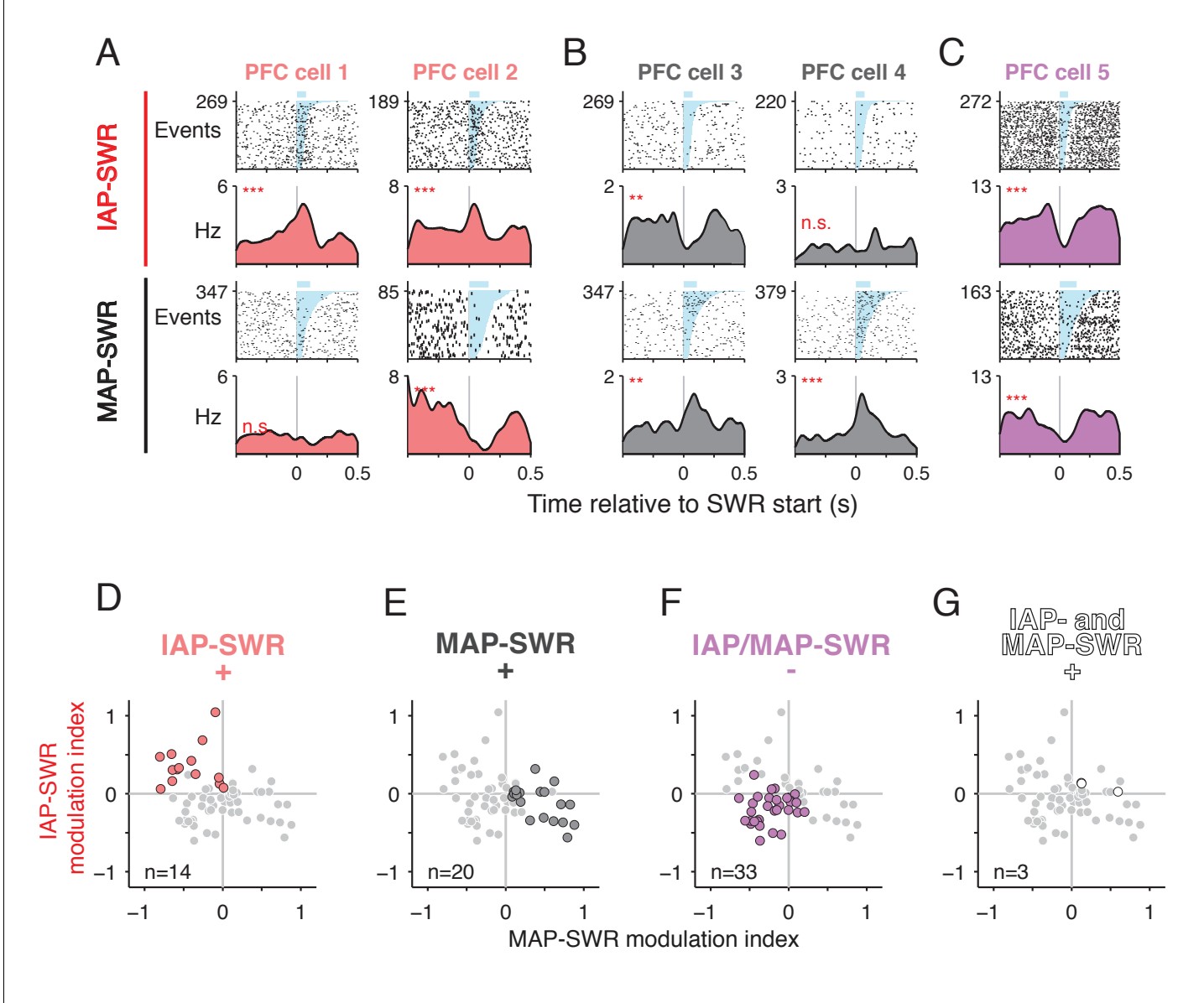

**Figure 5.** PFC modulation during SWRs distinguishes between MAP- and IAP- SWRs. (A-C) PFC spiking activity aligned to SWRs. Spike raster (first and third rows, number of SWR events shown on y axis) and mean firing rate (second and fourth rows) for each cell are shown. Upper two rows: firing aligned to IAP-SWRs. Lower two rows: firing aligned to MAP-SWRs. SWRs are sorted by length (shaded in cyan, mean duration shown by blue bar above raster). n.s. not significant, **p<0.005 and ***p<0.0005 using a circular permutation test (Materials and methods). (A) Two example cells showing a significant increase in firing during IAP-SWRs but no change (cell 1) or a significant decrease in firing (cell 2) during MAP-SWRs. (B) Two example cells showing a significant increase in firing around MAP-SWRs but no change (cell 3) or a significant decrease in firing (cell 4) during IAP-SWRs. (C) Example cell showing significant decreases in firing during both IAP-SWRs and MAP-SWRs. (D-G) PFC units showing significant changes in activity during SWRs (n = 70, gray circles) classified into one of four groups (colored circles), based on the significance of their modulation indices for IAP-SWRs (y axis) and MAP-SWRs (x axis). The gray circles are repeated in each panel and the colored circles highlight cells belonging to each group (cell count shown in each panel). (D) Significantly excited only during IAP-SWRs (IAP-SWR+) (observed = 14, expected = 18, Binomial test n.s.). See Materials and methods for calculation of expected values. (E) Significantly excited only during MAP-SWRs (MAP-SWR+) (observed = 20, expected = 18, Binomial test n.s.). (F) Significantly inhibited during either or both IAP-SWRs and MAP-SWRs (IAP/MAP-SWR-) (observed = 33, expected = 26, Binomial test: n.s.). (G) Significantly excited during both IAP-SWRs and MAP-SWRs (IAP- and MAP-SWR+) (observed = 3, expected = 9, Binomial test: p<0.05). Fewer than the expected number of cells was observed in this group.
DOI: https://doi.org/10.7554/eLife.27621.024

The following figure supplements are available for figure 5:

**Figure supplement 1.** PFC modulation patterns distinguish between the identity of IAP- and MAP-SWRs and not differences in SWR duration.

*Figure 5 continued on next page*

*Figure 5 continued*

DOI: https://doi.org/10.7554/eLife.27621.025

**Figure supplement 2.** PFC modulation patterns during MAP and IAP-SWRs.

DOI: https://doi.org/10.7554/eLife.27621.026

these differences in activity across the SWR types, we expressed the degree of change in the firing rate of individual PFC units as a pair of activity modulation indices, one for MAP-SWRs and the other for IAP-SWRs. An analysis of the distribution of these indices for significantly modulated cells showed that on a population level, the majority of PFC units had modulation indices corresponding to three main groups: excitation during IAP-SWRs (PFC IAP-SWR[+]), excitation during MAP-SWRs (PFC MAP-SWR[+]), or inhibition during either or both types of SWRs (PFC IAP/MAP-SWR[−]) (*Figure 5D–F*). Only 3 units showed significant excitation during both IAP- and MAP-SWRs (PFC IAP- and MAP-SWR[+]) (*Figure 5G*), which was fewer than expected by chance (expected = 9, Binomial test, observed vs. expected p<0.05).

These results demonstrate that a subset of PFC units shows selective excitation during either MAP-SWRs or IAP-SWRs but rarely both, while others, particularly those showing inhibition, do not distinguish between MAP-SWRs and IAP-SWRs. Differences in PFC responses could not be explained by differences in duration between IAP- and MAP-SWRs (*Figure 5—figure supplement 1A,C and D*). Thus, on a population level, PFC activity changes during SWRs can distinguish between MAP-SWRs and IAP-SWRs (*Figure 5—figure supplement 1A–B and D*), indicating that the distinction between immobility- and movement-associated reactivation events is also observed in PFC.

We also examined the modulation pattern of PFC units during the small fraction of SWRs that contained both MAPs and IAPs (MAP & IAP-SWRs). We found that PFC modulation patterns during MAP and IAP-SWRs were qualitatively similar to modulation patterns during both MAP-SWRs and IAP-SWRs (*Figure 5—figure supplement 2*). In a multiple regression of SWR modulation indices from 15 PFC units where we could examine the modulation indices of all three types of SWRs, we found the modulation indices of MAP-SWRs and IAP-SWRs together account for 73% of the explained variance of MAP & IAP-SWR modulation indices ($R^2$ = 0.73, p<0.0005). These results suggest that MAP & IAP-SWRs, although rare, represent a joint reactivation of both MAP-SWRs and IAP-SWRs representations in both CA1 and PFC.

## PFC SWR modulation pattern correlates with task activity

These findings demonstrate that PFC modulation patterns during SWRs differ depending on whether the hippocampus is reactivating a representation related to movement or immobility. If these coordinated activity changes across the hippocampus and PFC reflect a coherent reinstatement of distinct movement or immobility-associated experiences, we would expect the PFC modulation patterns during SWRs to be predictive of their activity pattern outside of SWRs during the task. To determine if that was the case, we examined the spiking pattern of PFC and CA1 cells in the transition period between movement and immobility periods (*Figure 6A–B* and *Figure 6—figure supplement 1*).

For PFC units belonging to each of the three modulation groups (*Figure 5D–F*), we found that individual unit firing patterns (*Figure 6A–B*, upper rows), average population activity (*Figure 6A–B*, lower rows) and pairwise similarity of PFC-CA1 firing patterns (*Figure 6C*) all indicated that SWR-modulation patterns were related to ongoing activity during behavior. Specifically, PFC IAP-SWR[+] units were most active at the onset of immobility and show spiking patterns most similar to CA1 IAPs. PFC MAP-SWR[+] cells were most active during movement before reward well entry and show spiking patterns most similar to CA1 MAPs. In contrast, PFC IAP/MAP-SWR[-] cells tended to show a gradual increase in spiking after reward well entry, a pattern different than that of IAPs or MAPs in CA1. Although the 3 PFC IAP- and MAP-SWR[+] units showed firing properties more similar to MAP-SWR[+], we hesitate to draw conclusions about their firing properties given the scarcity of this group and the lack of strong modulation to either SWR type (*Figure 6—figure supplement 1*).

Correspondence between spiking activity during SWRs and during ongoing experience was also seen when we examined patterns of spiking coactivity, which is a complementary measure of coactivity in time independent of task events such as reward well entry. We computed PFC-CA1 cell pair

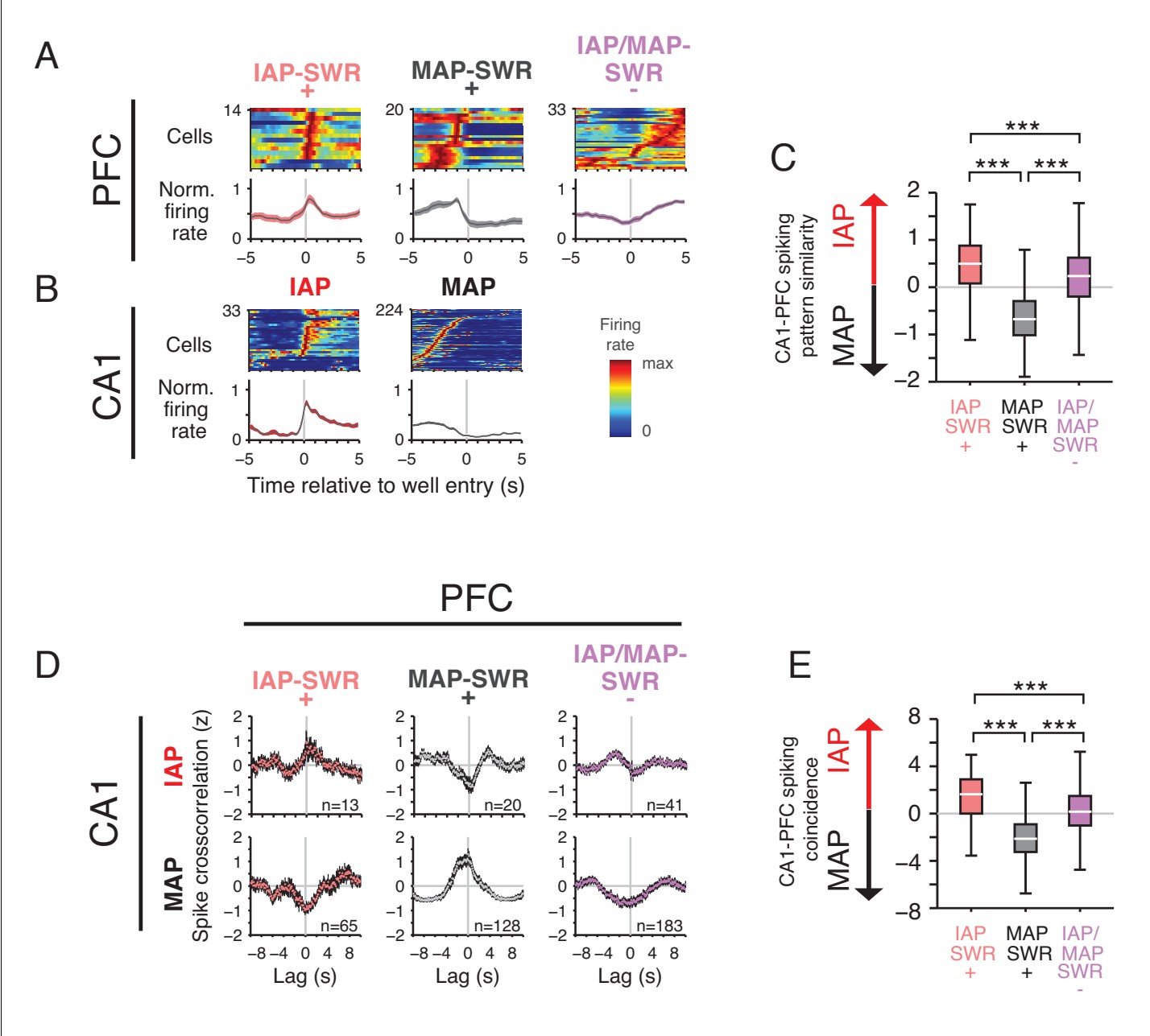

**Figure 6.** PFC SWR modulation reflects task firing properties. (**A**) Mean firing rate for significantly SWR modulated PFC units aligned to reward well entry. PFC units are grouped by their participation in IAP-SWRs and MAP-SWRs (**Figure 5D**). Top row: firing rate of each cell in each group normalized to maximum firing rate in the −5 s to 5 s time window. Bottom row: mean normalized firing rate (±SEM) for all cells in each group. (**B**) Mean firing rate for CA1 cells aligned to well entry. Top row: firing rate of each cell in each group normalized to maximum firing rate in the −5 s to 5 s time window. Bottom row: mean normalized firing rate (±SEM) for all cells in each group. (**C**) Firing pattern similarity index for each PFC cell group. The similarity index is the pairwise difference in correlation value between the firing pattern correlations for all possible PFC-CA1 IAP and all possible PFC-CA1 MAP cell pairs. A value greater than 0 indicates similarity to IAP firing whereas a value less than 0 indicates similarity to MAP firing. Wilcoxon rank-sum test: ***p<$10^{-4}$. Pairwise comparisons: IAP-SWR[+] n=1082067, MAP-SWR[+] n=1921500 and IAP/MAP-SWR[-] n = 3606768. (**D**) Normalized population spike cross-correlation for PFC-CA1 cell pairs for all times outside of SWRs (mean ±SEM, 100 ms bins). Top row: PFC- CA1 IAP pairs. Bottom row: PFC-CA1 MAP cell pairs. Number of cell pairs is indicated. Lag refers to time of CA1 spiking relative to each reference PFC spike. (**E**) Spiking coincidence index for each PFC cell group. The spiking coincidence index is the pairwise difference in spike cross-correlation (100 ms window centered at 0 s lag) between PFC-CA1 IAP (D: top row) and PFC-CA1 MAP (D: bottom row) cell pairs. A value greater than 0 indicates higher spiking coincidence with IAPs whereas a value less than 0 indicates higher spiking coincidence with MAPs. Wilcoxon rank-sum test: ***p<$10^{-4}$. Pairwise comparisons: IAP-SWR[+] n=845, MAP-SWR[+] n=2560, IAP/MAP-SWR[-] n = 7503.

DOI: https://doi.org/10.7554/eLife.27621.027

*Figure 6 continued on next page*

*Figure 6 continued*

The following figure supplements are available for figure 6:

**Figure supplement 1.** PFC-CA1 activity relationships for IAP- and MAP-SWR[+] cells.

DOI: https://doi.org/10.7554/eLife.27621.028

**Figure supplement 2.** Spatial and temporal proximity control for PFC activity.

DOI: https://doi.org/10.7554/eLife.27621.029

spike cross-correlations for all spikes outside of SWRs (*Figure 6D*) and found that PFC IAP-SWR[+] cells were more coactive with CA1 IAPs than CA1 MAPs. Conversely, PFC MAP-SWR[+] cells were more coactive with CA1 MAPs than CA1 IAPs. PFC IAP/MAP-SWR[-] cells were least coactive with either type of CA1 place cell. These effects could not be explained by differences in spatial or temporal proximity of spiking between different PFC cell pairs (*Figure 6—figure supplement 2*) but were consistent with the pattern of coordinated activity between PFC and CA1 outside of SWRs being preserved in the network and reactivated during SWRs.

## Discussion

We found CA1 place cells active during movement or immobility form a continuous spatial representation of ongoing experience. Despite this continuity, ~93% of SWR reactivation events contained either movement or immobility-associated spatial representations but not a mixture of the two. Thus, the hippocampus can express distinct reactivation of events even when those events are represented continuously in time during ongoing experience. The distinction between movement and immobility reactivations was found also in PFC, where modulation of PFC activity during SWRs distinguishes between movement- and immobility-associated content in the hippocampus. PFC modulation during SWRs was correlated with their activity pattern outside of SWRs during the task. This demonstrates the engagement of the hippocampal-cortical network during reactivation of both immobility- and movement-associated representations.

### A continuous spatial representation in CA1

We found that CA1 maintained a continuous spatial representation across different behavioral states. During the transition period between movement and immobility, the spiking of MAPs (cells that participate in movement related spatial representations) and IAPs (cells that participate in immobility related spatial representations) overlapped in time on the time scale of seconds. Interestingly, although IAPs were predominantly active during immobility when the theta oscillation is typically absent, they showed theta modulation during the higher speeds during the transition period, suggesting IAP entrainment to both theta and N-waves.

When an animal remains immobile, its experience continues in time but does not change in spatial location. In the hippocampus, activity consistent with representations of elapsed time have been found, where 'time cells' tile time while an animal remains immobile during a waiting period (*MacDonald et al., 2011*). In our tasks the animal was not required to explicitly time or anticipate the length of immobility periods at reward well locations but was instead allowed to initiate movement bouts voluntarily. This could perhaps explain the lack of sequential activity among IAPs when we examined the firing patterns of multiple IAPs aligned to the onset of immobility (*Figure 6B*). However, it is possible that ensembles of IAPs may be recruited to form temporally patterned activity in tasks that require keeping track of information over time during immobility.

The IAPs we describe here share similar properties to N-wave coupled cells that participate in immobility spatial coding (*Kay et al., 2016*) or have spatially selective activity during immobile waiting periods (*McKenzie et al., 2013*). They are distinct from previously described CA1 units that are active near goal locations, which have been proposed to represent learned expectation of reward at specific locations (*Gothard et al., 1996*; *Hok et al., 2007*; *Kobayashi et al., 2003*). First, in both of our tasks, the animal could obtain reward at multiple locations. If IAPs represented reward expectation in our tasks, we would expect their activity to be associated with multiple locations where reward was expected. That was not the case: we found that IAPs tended to be spatially selective for one location (*Figure 1C*). Secondly, the spikes of IAPs occurred mostly when the animal was immobile and their activity spanned less space compared with MAPs (*Figure 1D–E*), which indicates the

main spatially selective activity for IAPs was restricted to periods of immobility. This contrasts with previously described goal-related place cells that have dominant place fields away from the goal location (*Hok et al., 2007*). Lastly, the foraging task involved multiple changes in spatial reward contingency, where the locations of rewards shifted (*Figure 1—figure supplement 1A*). If IAPs encoded learned spatial reward expectation, IAP activity should follow changes in reward location, which has been observed previously (*Gothard et al., 1996*; *Kobayashi et al., 1997*; *Kobayashi et al., 2003*). Again, we found IAP activity remained stable and did not shift to newly rewarded locations (*Figure 1—figure supplement 3A*). Thus, IAPs are functionally distinct from previously described goal-related firing in CA1.

## Distinct reactivation of immobility-associated representations

Our results show that SWR reactivation of immobility-associated representations is largely distinct from reactivations of movement-associated representations, despite the smooth transitions between these representations during behavior. Previous studies have classified SWRs into distinct classes based on unique LFP signatures (*Ramirez-Villegas et al., 2015*) and showed that information extracted from the LFP can be used to predict the content of trajectory replay events (*Taxidis et al., 2015*). Consistent with those observations, we found SWRs reactivating immobility-associated experiences tended to be shorter in duration and lower in amplitude than SWRs reactivating movement-associated experiences. Given the durations of SWRs are correlated with the spatial extent of reactivated spatial representations (*Davidson et al., 2009*), the difference in durations of SWRs reactivating movement- or immobility-associated experiences is consistent with the spatial coverage of their respective ongoing representations.

We also found SWRs containing immobility representations were accompanied by concurrent and coordinated changes in the activity of subsets of PFC neurons. The PFC units that showed excitation during IAP-SWRs also showed similar firing coactivity to CA1 IAPs outside of SWRs during the task, with a peak in activity immediately upon reward well entry. Therefore, a subset of SWRs marks the coordinated reinstatement of multimodal representations of immobility-associated experiences across different brain areas. Our results establish that there are distinct, largely mutually exclusive classes of SWRs that reinstate hippocampal and PFC representations associated with experience during different behavioral states.

The highly selective PFC modulation patterns that accompany reactivated hippocampal representations of either movement- or immobility-associated experiences suggest this distinction has important roles that extend beyond the hippocampus. Movement and immobility related modulation patterns in PFC during SWRs may reflect the concurrent reactivation of non-spatial aspects of experience associated with immobility, such as representations of actions and their consequences (*Euston et al., 2012*; *Sul et al., 2010*). These reactivations may be required for local PFC processes that strengthen specific cortical representations of experience in memory or may reflect access to locally stored information for computations involved in planning upcoming choices (*Yu and Frank, 2015*).

The independent reactivation of movement- and immobility-associated representations could also reflect the distinct roles of these experiences for guiding future choices. For rodents, exploratory behavior is characterized by alternating bouts of movement and immobility (*Drai et al., 2001*; *Whishaw and Kolb, 2005*), which are associated with distinct experiences. Navigational planning represents one of the many challenges faced by foraging animals. In natural environments with complex terrains and spatial topologies, behaviorally significant locations are distributed in space and can be reached by many paths (*Calhoun, 1963*; *Telle, 1966*). Memory of experience during movement may assist in selecting routes to specific locations of interest in the environment. An animal may need memories of experiences that occurred at these locations to choose what to do once it is there. Thus, distinct memories for immobility-associated experiences may provide context and guide decisions specific to certain locations in order to obtain reward or avoid punishment.

## A Hebbian mechanism for separating movement and immobility representations

What mechanism could underlie the independent reactivation of movement and immobility spatial representations despite the smooth transition during ongoing experience? Previous work has

demonstrated that the temporal spiking patterns of place cells during movement are predictive of their association during subsequent SWR reactivation (*O'Neill et al., 2008*). These results suggested a model where coincident spiking between a pair of place cells, most likely in upstream area CA3 where there are extensive local recurrent collaterals (*MacVicar and Dudek, 1980*; *Miles and Wong, 1983*; *1986*), strengthens their association, thereby linking individual place fields for joint reactivation during subsequent SWRs. At the same time, non-coincident spiking of place cells with place fields that are separated in space and time would decrease the strength of their association, reducing the likelihood that they will be incorporated into the same reactivated representation (*Figure 7A,B*).

In our tasks, although MAPs and IAPs showed coincident spikes during the transition period between movement and immobility that could lead to the strengthening of their associations, the proportion of coincident spikes in a short time window compared with spikes over a longer period was significantly less than for pairs of MAPs (*Figure 2—figure supplement 2*). This difference was due to IAP activity that continues for many seconds during immobility, which lead to a greater number of non-coincident spikes between MAP and IAP pairs and could weaken their association. In contrast, during immobility, groups of IAPs that have local place fields were active, which would drive increased association between pairs of IAPs (*Figure 2—figure supplement 2*). As a result, we expect

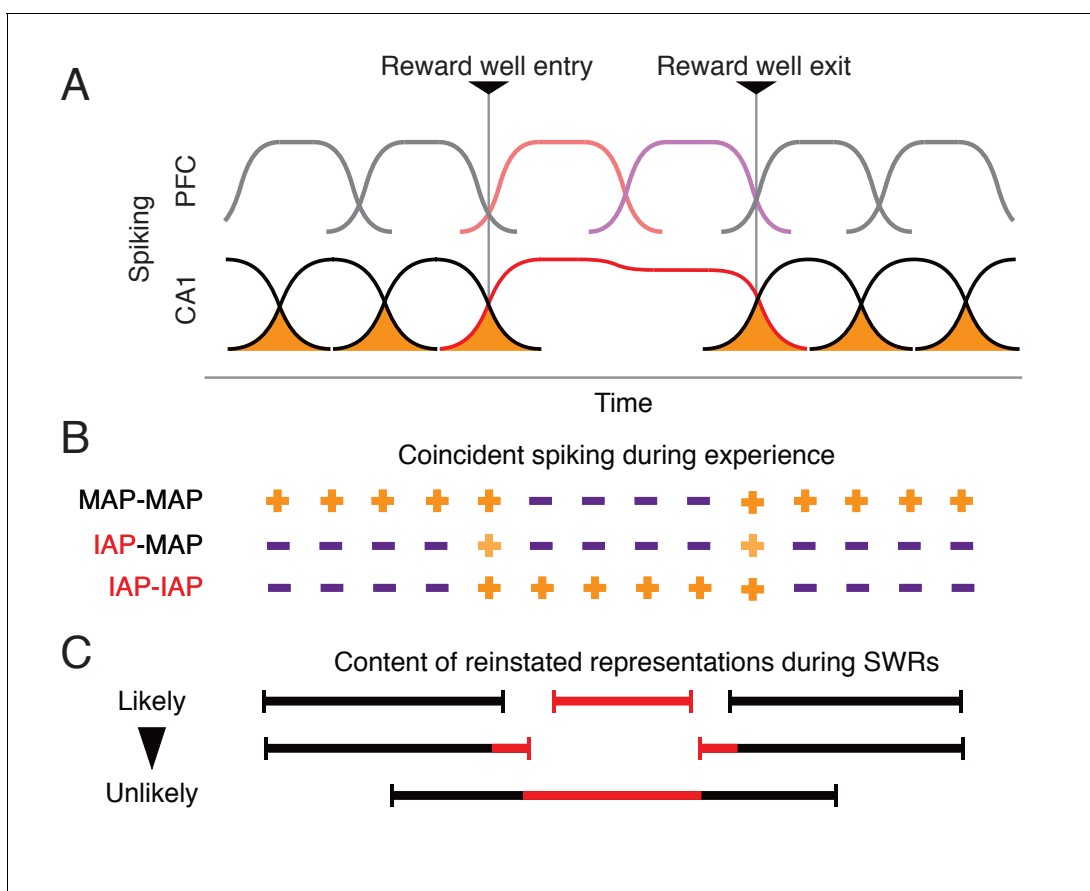

**Figure 7.** A Hebbian mechanism for discretizing continuous representations of experience. (**A**) Schematic of PFC and CA1 activity around reward well entry and exit. The activity of each cell over time is shown by a curve. The overlap in activity between CA1 cells is shaded orange. CA1 MAPs shown in black. CA1 IAPs shown in red. Corresponding PFC activity follows color scheme in *Figure 5*. (**B**) Schematic indicating contribution of spiking coactivity at different points in time to the strengthening of associations for MAPs and IAP pairs. Orange plus symbol indicates associations between cell pairs. The lighter shade of orange for IAP-MAP pairs indicates weaker association compared with other cell pairs. Purple minus symbol indicates weakening of associations. (**C**) Expected representations reactivated during SWRs based on associations in (**B**). Each line represents one type of reinstated representation. Black indicates movement-associated representations and red indicates immobility-associated representations.
DOI: https://doi.org/10.7554/eLife.27621.030

increases in association within ensembles of MAPs and within ensembles IAPs, but less strengthening or even weakening of association between MAPs and IAPs (*Figure 7A–B*).

A prediction of this hypothesis is that during SWR reactivation events, which reflect associations formed between place cells during ongoing experience, we will see a predominance of events with either MAPs or IAPs but not both (*Figure 7C*, top). A less frequently observed type of SWRs will include both types of place cells, where reactivation events would have an initiation bias for the immobility representation (*Figure 7C*, middle). Least likely are reactivations where the immobility-associated representations are sandwiched between two movement representations (*Figure 7C*, bottom). Our data is consistent with all these predictions. We saw the majority of SWRs contained either MAPs or IAPs. In the minority of SWRs with both MAPs and IAPs, we found IAPs tended to spike earlier in SWRs compared with MAPs, which is consistent with locations of immobility being found mostly at the start rather than distributed within putative trajectory reactivation events. We are also unaware of any reports of reactivation events that contain representations of a trajectory that leads to a location of immobility followed by a trajectory that leads away from the location.

More broadly, we suggest this Hebbian mechanism could support the transformation of adjacent periods of ongoing experience into separate memories. Findings from human studies suggest the division of experience into episodes is an ongoing cognitive process that is shared across species. Humans can reliably report boundaries that separate perceived events and episodes in ongoing experience (*Newtson, 1973*; *Zacks et al., 2001b*) and brain activity shows marked changes in activity coinciding with progression across these perceived boundaries (*Speer et al., 2007*; *Sridharan et al., 2007*; *Whitney et al., 2009*; *Zacks et al., 2001a*). We suggest temporally adjacent events represented by the activity of cells with receptive fields that also overlap in time, on the scale of seconds, may become unique representations if the number of spikes that overlap, within small time intervals, during the transition between the two events are small compared with the overall number of spikes fired individually during each event. This could then lead to the creation of separate episodes of experience in memory.

## Materials and methods

### Animals

Eleven male Long Evans rats (500–700 g, 4–9 months old) were used for these studies. Six of the animals performed the W-shaped alternation task and data from those animals contributed to previous publications (*Karlsson and Frank, 2009*; *Kay et al., 2016*). Five animals performed the foraging task. All experiments were conducted in accordance with University of California San Francisco Institutional Animal Care and Use Committee and US National Institutes of Health guidelines.

### Implant coordinates

Animals that performed the W-shaped alternation task were implanted with tetrodes targeting the hippocampus. Coordinates have been described previously (*Kay et al., 2016*). Animals that performed the foraging task were implanted with tetrodes targeting both dorsal CA1 and dorsal PFC (anterior cingulate cortex and dorsal prelimbic cortex) (*Figure 1—figure supplement 2*).

CA1: seven independently movable tetrodes targeted the right dorsal CA1 at AP: −3.8 mm and ML: 2.2 mm.

PFC: 14–21 independently movable tetrodes located in one cannula angled at 20 degrees toward the midline targeted towards the right dorsal PFC at AP: +2.2 mm, ML +1.5 mm. Tetrodes were lowered gradually to a DV depth between 1.88 mm to 2.72 mm depending on the AP and ML coordinates of each tetrode.

After surgery, tetrodes were adjusted every 2 days to reach the desired DV coordinate (PFC) or guided by LFP and spiking patterns to reach the cell layer (CA1). After the start of the experiments, tetrodes were adjusted at the end of the day in small increments (typically ~30 μm) to improve cell isolation. Tetrode locations were verified histologically.

### Drive design

Recording drives were custom designed and 3D printed (PolyJetHD Blue, Stratasys Ltd.) to house a maximum of 28 individually movable tetrodes. Tetrodes (Ni-Cr, California Fine Wire Company) were

gold plated to reach an impedance of 250kOhm at 1 kHz and were grouped in cannulas to target the PFC and/or CA1.

## Pre-training

Training of W-shaped alternation track has been described previously (*Kay et al., 2016*). For the foraging task, animals were initially trained to run on a 1 m long linear track for reward (evaporated milk plus 5% sucrose, Carnation). The reward (100–300 µl at 20 ml/min) was delivered using a syringe pump (NE-500 OEM, New Era Pump Systems Inc.). The pump was triggered immediately (2 animals) or after a 1 s delay (3 animals) after the animal's nose crossed an infrared beam on the reward delivery device (reward well) located at each end of the track.

## Behavior tasks

We analyzed data from two memory guided spatial navigation tasks. Both tasks involve interleaved periods of movement between reward locations and immobility at reward locations. Our goal was to examine movement- and immobility-related representations in environments with different topologies and task rules.

Foraging task: ~21 days after surgery, animals were introduced to the foraging task (*Figure 1—figure supplement 1A*). For each session, two of the four possible reward locations were chosen to deliver reward. The rat had to find those two locations and visit them in alternation to receive reward. The rewarded locations remained the same for all three daily sessions for days 1–8 but were changed between days to cover all possible combinations. The reward locations then changed between the three daily sessions for the next 2–4 days before switching within and between every session for the next 5–9 days. For the last 3–4 days, all three sessions had the same reward well locations but in one session, the animal had to navigate around a barrier on one of the segments of the environment. Changes in spatial reward contingencies within and between sessions were not explicitly signaled and the rat had to discover the new reward locations by trial and error. Reward in the task was delivered using the same system for pre-training (see Pre-training) and the contingency change was computer programmed. Data from 4, 9, 9, 11 and 12 days were analyzed for each of the five animals respectively. Each day consisted of between 2–3 task sessions of 15–45 min each. The task sessions were interleaved with rest sessions of 20–60 min in a sleep box located away from the track.

W-shaped alternation task (*Frank et al., 2000*; *Jadhav et al., 2012*; *Kim and Frank, 2009*): ~14 days after surgery, animals were introduced to the W-shaped track. Animals were required to visit each reward well in an alternating order to receive reward (*Figure 1—figure supplement 1A*).

## Histology

To mark the location of recording sites, electrolytic lesions were made by passing current through each tetrode (30 µA, 3 s) and animals were perfused 12–24 hr later with paraformaldehyde (4% in PBS). The brain was fixed for 24 hr at room temperature and then cryoprotected with sucrose (30% in PBS at 4°C). Coronal sections (50 µm) were stained with Cresyl Violet to identify sites of electrolytic lesions.

## Recording

The NSpike data acquisition system (LMF and J. MacArthur, Harvard Instrumentation Design Laboratory) was used for data collection. Experiments were carried out in dim lighting and video of behavior was recorded at 30 Hz. A custom designed circular infrared LED array was mounted on the headstage amplifier for tracking the animal's position in the environment. The array consists of a large group of LEDs to indicate the front and a small LED mounted on a extender to signal the rear or the array. LFP (filtered from 0.5 to 400 Hz) on each tetrode was recorded at a sampling rate of 1.5 kHz. Spiking activity (filtered from 600 to 6000 Hz or 300–6000 Hz) was recorded from each tetrode channel at a sampling rate of 30 kHz. A tetrode located in corpus callosum was used as a reference for LFP and spike detection for hippocampal tetrodes. A tetrode located in PFC but did not detect spikes was used as reference for PFC tetrodes.

## Data analysis

Putative neurons (units) were identified by manual clustering of spiking data from channels of each tetrode based on peak amplitude, spike width and wave-form principal components (MatClust, M.P. K.). Only stable and well-isolated units were used for further analysis.

## Behavioral state definition

The animal's position was determined as the centroid of the front and back diodes from the LED array using a semiautomated analysis of the video. Movement was defined as periods when the speed of the tracked position was >4 cm/s. Immobility was defined as periods when the speed of the tracked position was <4 cm/s and almost all immobility periods occurred at reward well locations (*Kay et al., 2016*).

## Cell selection

CA1: Putative pyramidal neurons were separated from fast spiking interneurons based on spike width (<0.4 ms) and mean firing rate (<10 Hz). Only cells with >200 spikes across all sessions on a given day were included. IAPs and MAPs were defined as cells with or without a place field (mean firing rate $\geq$3 Hz), respectively, during periods of immobility at a reward well, excluding spikes during SWRs. We calculated two other parameters, the median speed per spike and the place coverage, which show IAPs and MAPs have distinct spatial firing properties.

PFC: We recorded from 712 PFC units. For further analysis, we selected cells (n = 112) that were simultaneously recorded with CA1 IAPs and MAPs and had a mean firing rate >1 Hz during movement or immobility periods or both.

## Well specificity

Well specificity was calculated as previously described (*Kay et al., 2016*). The index was calculated by normalizing the mean firing rate of the IAP at each well to the maximum. Each firing rate is then converted into a unit vector where each vector is equally separated on a unit circle. For example, for the four well foraging task, each vector is separated by 90 degrees. The vectors are then summed and the magnitude the resulting vector is the specificity index. A value of 1 indicates the cell is only active at one well while a value of 0 indicates the cell is equally active at all wells.

## SWR detection

Our SWR detection approach was modified from previous work (*Csicsvari et al., 1999*; *Kay et al., 2016*). The raw CA1 LFP was referenced to an electrode in corpus callosum and then filtered (150–250 Hz) to isolate the SWR band. The SWR envelope was then obtained using the Hilbert transform and then smoothed by convolving with a Gaussian kernel ($\sigma$ = 4 ms). A consensus SWR envelope was calculated by taking the median of the envelopes across all available tetrodes. Only days where at least three tetrodes were in or near the CA1 cell layer were used for the analysis.

Standard methods for identifying SWRs involve setting a fixed detection threshold for events (e.g. 3 SD above mean) (*Diba and Buzsáki, 2007*; *Jackson et al., 2006*; *Karlsson and Frank, 2009*; *Lee and Wilson, 2002*; *O'Neill et al., 2006*). We noted that this method could yield somewhat different sets of events across days because the characteristics of the consensus envelope can be variable across animals, days and behavior states. Thus, using a fixed detection threshold to define SWRs could include spurious events on days with higher noise level and exclude genuine events on days with a lower noise level. Furthermore, this cutoff is typically arbitrary, therefore prompting us to develop a more principled approach to distinguish events from noise.

To avoid the problem of arbitrary thresholds and differences in noise distributions across days, we developed a SWR identification approach that defines a threshold based on the distribution of the consensus envelope power for a given day (*Figure 3—figure supplement 1*). We first normalized the consensus envelope power to times during immobility (speed <4 cm/s). We assumed that the full distribution of the envelope power was the result of a lower power noise component and a higher power signal component that contributes to the long tail. We approximated the noise distribution by finding the mode for the histogrammed consensus envelope power and creating a symmetric distribution by mirroring the histogram below the peak around the mode. We then used a cutoff (99.99th percentile) in envelope power based on this empirically estimated noise distribution

as the detection threshold for SWRs. This method allowed us to capture the distinct noise distributions for immobility periods for each day. SWRs were then detected when the power of the consensus envelope exceeded the detection threshold for a minimum of 20 ms. The starts and ends of the events were set to times when the envelope power returns to the mean. Only SWR events occurring at speed <4 cm/s were included in our analyses.

## Occupancy normalized firing maps

The environment was first divided into 2 cm square bins. The occupancy-normalized rate was calculated by dividing the number of spikes by the occupancy of the animal per bin and smoothing with a 2-dimensional symmetric Gaussian kernel (σ = 2 cm and 12 cm spatial extent).

## Cross-correlations and auto-correlations

For CA1, cross-correlations (*Figure 2*, *Figure 2—figure supplement 1E–H*) or auto-correlation (*Figure 2—figure supplement 1C–D*) were histogrammed in 10 ms time bins over a time window (0.5 s, 1 s or 6.6 s) depending on the time interval of interest. Only cell pair with >100 coincident spiking events within the time window of interest were included. The cross-correlogram was normalized by dividing the value of each bin by the sum of all bins. The population median for each bin was then calculated.

## SWR counts

We classified SWRs based on their spiking content as MAP-SWRs (containing MAP spikes only), IAP-SWRs (contain IAP spikes only) or MAP & IAP-SWRs (containing both MAP and IAP spikes). The expected number of MAP & IAP-SWRs is calculated as the joint probability of observing SWRs with MAPs and SWRs with IAPs, under the assumption of independence of either event. The observed probability is compared with the expected using a Chi-square test with 1 degree of freedom.

## SWR coactivity

SWR coactivity for a pair of cells was calculated using a previously defined measure which normalizes the observed probability of both cells participating in the same SWR by the expected probability of observing both cells participating in the same SWR (*Cheng and Frank, 2008*). For each cell, its spike count during a SWR was first binarized, where the cell was either participating or not participating in a SWR. The proportion of all SWR events where both cells were firing was the observed coactivity. The expected coactivity was calculated by permuting the participation of each cell across all SWRs. This was performed 5000 times and to generate a distribution of the expected proportion of SWRs with both cells active. The observed proportion was converted to a z-score by subtracting the mean and dividing by the standard deviation of the expected distribution. This method takes into account the differences in SWR participation for each cell in the pair.

We chose this method to examine the content of SWRs instead of trajectory decoding methods because the latter enriches for SWRs that contain multiple place cells and decode to trajectories matching predefined templates, which often only consists of place cells with activity during movement (e.g. MAPs). However, SWRs that do not meet these criteria, either with too few participating place cells or do not decode to a meaningful trajectory, are classified as non-replay events and excluded from further analysis. The pairwise coactivity method captures structured SWR activity that has been related to activity during recent experiences (*O'Neill et al., 2006*; *Singer and Frank, 2009*) but does not impose conditions that select for certain types of content.

## Spectrograms

We used the Chronux toolbox (*Mitra and Bokil, 2008*) (chronux.org) to calculate spectral power. The broadband CA1 LFP was referenced to ground. The spectrogram was calculated in 100 ms sliding windows with a 10 ms step size and normalized by subtracting the mean and dividing by the standard deviation from all times during the session. We then calculated the mean from all 500 ms windows centered on the start each SWR. To quantify the spectral power for SWRs, we calculated the mean spectral power for the first 100 ms after the onset of each SWR. We then compared the mean power between 150–250 Hz in this window.

## PFC SWR modulation index

The modulation index is the change in firing during SWRs relative to the surrounding mean firing rate and is calculated for each cell and each type of SWR as:

$$I_{SWR} = \frac{r_{SWR} - r_{1s\ window}}{r_{1s\ window}}$$ where $I_{SWR}$ is the SWR modulation index, $r_{SWR}$ is mean firing rate during SWRs and $r_{1s\ window}$ is the mean firing rate in the 1 s window centered on SWRs. The length of time used for calculating $r_{SWR}$ is the mean durations of all SWR events for a given type of SWR (*Figure 5A–C*, cyan bar above spike raster).

The significance of modulation was calculated as describe previously (*Jadhav et al., 2016*). For a given type of SWR, we first generated a perievent time histogram (PETH) for all events aligned to the start of SWRs for the observed data. We then generated a control dataset by circularly permuting the spike times for each SWR event, such that all spikes around one SWR event were circularly shifted by the same amount but this amount varied between SWR events. From this control dataset we then generated a PETH. This was repeated 1000 times. Next we calculated the squared deviation of the observed PETH from the mean of the 1000 control PETHs for the average duration of SWRs for the given type of SWR. We then compared the squared deviation of each of the 1000 control PETHs to the mean of all 1000 control PETHs. The significance value was the fraction of 1000 control PETH deviations that are larger than the observed PETH deviation.

As a control to evaluate the difference in PFC activity changes during IAP- and MAP-SWRs, the identity of the SWR was permuted before recalculating the SWR modulation index. As a control for the difference in duration between IAP- and MAP-SWRs, SWR events for each PFC cell were resampled to match the duration distribution of IAP- and MAP-SWR groups before recalculating the SWR modulation index.

## PFC cell classification

For each PFC unit, we calculated its SWR modulation index for each type of SWR. For the PFC units that showed significant modulation to either type of SWR, we classified them into 4 groups based on the following criteria: significant excitation only during IAP-SWRs (units showing IAP-SWR[+] MAP-SWR[−] or IAP-SWR[+] MAP-SWR[n.s.]), significant excitation only during MAP-SWRs (MAP-SWR[+] IAP-SWR[−] or MAP-SWR[+] IAP-SWR[n.s.]), significant inhibition during either or both IAP/MAP-SWRs (MAP-SWR[n.s.] IAP-SWR[−], MAP-SWR[−] IAP-SWR[n.s.] or MAP-SWR[−] IAP-SWR[−]) and significant excitation during both IAP/MAP-SWRs (IAP-SWR[+] MAP-SWR[+]). The expected number of units for each of the 4 groups was calculated under the assumption that the 70 PFC units are randomly distributed among the possible combinations of modulation significance (n = 8, listed in brackets). For example, the expected number of units with significant excitation only during IAP-SWRs, which has two possible combinations of modulation significance, is 18 (2 × 70/8). The significance of the difference between observed and expected values was calculated with a Binomial test.

## PFC and CA1 spiking activity

PFC and CA1 spiking was aligned to reward well entry, as measured by an infrared beam break at the reward well. The mean instantaneous firing rate for a 10 s window centered on well entry was calculated for all well entries and divided by the maximum rate in this time window.

The spiking pattern correlation between PFC and CA1 cell groups is the pairwise Pearson's correlation of reward well entry aligned spiking pattern of all PFC-CA1 pairs for each comparison group. The spiking pattern similarity index for each PFC cell group was defined as the pairwise difference between the spike pattern correlation to CA1 IAPs and MAPs. This is an indication of whether the spiking pattern of a PFC cell group is more similar to the spiking of IAPs (>0) or MAPs (<0).

Cross-correlations between CA1-PFC unit pairs (*Figure 6*) were calculated in 100 ms bins with a window of 20 s (*Perkel et al., 1967*). For each pair, the cross-correlation was normalized by subtracting the mean and then dividing by the standard deviation of all bins in the 20 s period. This normalization method preserves the shape of the cross-correlation and allows for comparison across cell pairs.

The spiking coincidence measure for each PFC group was defined as the pairwise difference between the IAP and MAP spike cross-correlation values in a 100 ms time bin centered at 0 s lag. This is an indication of whether the spiking of cells in each PFC group is more coincident with IAP (>0) or MAPs (<0).

## Acknowledgements

We thank G Rothschild, D Kastner, E Anderson, B Mensch, E Phillips, C Geaghan-Breiner, A Gillespie, T Davison and H Joo for comments on the manuscript. This work was supported by a Jane Coffin Childs Memorial Fund for Biomedical Research postdoctoral fellowship (JYY), the Howard Hughes Medical Institute, NIH RO1MH105174, NIH R01MH097084 and University of California Office of the President Lab Fees Award #LF-12–237680 (LMF).

## Additional information

### Funding

| Funder | Grant reference number | Author |
|---|---|---|
| National Institute of Mental Health | RO1MH105174 | Jai Y Yu<br>Kenneth Kay<br>Daniel F Liu<br>Marielena Sosa<br>Jason E Chung<br>Loren M Frank |
| Jane Coffin Childs Memorial Fund for Medical Research | | Jai Y Yu |
| University of California | LF-12-237680 | Loren M Frank |
| Howard Hughes Medical Institute | | Jai Y Yu<br>Kenneth Kay<br>Loren M Frank |
| National Institute of Mental Health | R01MH097084 | Loren M Frank |

The funders had no role in study design, data collection and interpretation, or the decision to submit the work for publication.

### Author contributions

Jai Y Yu, Conceptualization, Formal analysis, Funding acquisition, Methodology, Writing—original draft, Writing—review and editing; Kenneth Kay, Daniel F Liu, Marielena Sosa, Jason E Chung, Writing—review and editing, Data acquisition; Irene Grossrubatscher, Adrianna Loback, Mattias P Karlsson, Margaret C Larkin, Data acquisition; Loren M Frank, Conceptualization, Supervision, Funding acquisition, Methodology, Writing—original draft, Project administration, Writing—review and editing

### Author ORCIDs

Jai Y Yu http://orcid.org/0000-0003-0628-2531
Loren M Frank http://orcid.org/0000-0002-1752-5677

### Ethics

Animal experimentation: All experiments were conducted in accordance with University of California San Francisco Institutional Animal Care and Use Committee and US National Institutes of Health guidelines. The protocol was approved by the Institutional Animal Care and Use Committee, approval number AN110101-03B. All surgical procedures were performed under anesthesia and every effort was made to minimize suffering.

### Decision letter and Author response

Decision letter https://doi.org/10.7554/eLife.27621.035
Author response https://doi.org/10.7554/eLife.27621.036

## Additional files

### Major datasets

The following dataset was generated:

| Author(s) | Year | Dataset title | Dataset URL | Database, license, and accessibility information |
|---|---|---|---|---|
| Yu JY, Liu DF, Grossrubatscher I, Frank LM | 2017 | Simultaneous extracellular recordings from hippocampal area CA1 and dorsal medial prefrontal cortex from rats performing a foraging task | http://dx.doi.org/10.6080/K0H41PK8 | Publicly available at the Collaborative Research in Computational Neuroscience data sharing website (http://crcns.org/) |

The following previously published dataset was used:

| Author(s) | Year | Dataset title | Dataset URL | Database, license, and accessibility information |
|---|---|---|---|---|
| Karlsson MP, Carr M, Frank LM | 2015 | Simultaneous extracellular recordings from hippocampal areas CA1 and CA3 (or MEC and CA1) from rats performing an alternation task in two W-shaped tracks that are geometrically identically but visually distinct | http://dx.doi.org/10.6080/K0NK3BZJ | Publicly available at the Collaborative Research in Computational Neuroscience data sharing website (http://crcns.org/) |

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
