## [Decision Letter]

Thank you for submitting your article "Distinct hippocampal-cortical memory representations for experiences associated with movement versus immobility" for consideration by *eLife*. Your article has been favorably evaluated by Eve Marder (Senior Editor) and three reviewers, one of whom is a member of our Board of Reviewing Editors. The following individual involved in review of your submission has agreed to reveal his identity: Jozsef Csicsvari (Reviewer #3).

The reviewers have discussed the reviews with one another and the Reviewing Editor has drafted this decision to help you prepare a revised submission.

The reviewers agree that the results concerning replay of immobility-related place cell firing and related prefrontal activity are potentially interesting and significant. However, they also identify some significant issues concerning whether the results are specific to the IAP/MAP distinction you make, or have a more general explanation that would be consistent with previous findings. These issues would need to be addressed in a revised manuscript, see below for details.

*Reviewer #1:*

The authors record place cells active during mobility (MAPs) and those active during immobility at reward locations (IAPs). They find a smooth transition between these types of cells when the animal enters or leaves a reward location.

The main findings concern 'reactivation events' during immobility and sharp-wave/ripples (SWRs), in that both types of cell are reactivated, but seemingly reactivation events are unlikely to contain both IAPs and MAPs. A second finding is that the patterns of firing of neurons in PFC are different during IAP versus MAP-related SWRs.

The paper is presented as showing that mobility and immobility-related events are reactivated separately, with the implication that different behavioral states are treated separately when forming and reactivating memories.

As I read the paper several basic questions came to mind that needed to be resolved before the results could be interpreted or evaluated properly (particularly Qs 1 and 2 below). These may be answered within the paper, but if so not in such an obvious way that I could tell.

1) Given that reactivation of firing during SWRs is most commonly thought of in terms of 'replay' of experience, and such 'replay events' correspond to experiential trajectories, one would expect the probability of reactivation of a pair of cells to depend on the spatial proximity of their firing fields. Does the IAP/MAP separation effect here simply reflect the increased spatial proximity of the firing fields of cells that fire during a run to a reward or at the reward location relative to mixed pairs across both populations (i.e. nothing to do with behavioral state, which just happens to be confounded with spatial proximity)? To answer this question requires plots of probability of co-reactivation versus the spatial distance between firing fields for all three types of pairs (MAP-MAP, MAP-IAP, IAP-IAP).

2) Given that reactivation of firing during SWRs is most commonly thought of in terms of 'replay' of experience, and such 'replay events' correspond to experiential trajectories, one would expect the probability of reactivation of a pair of cells to depend on the temporal proximity of their firing during experience. Does the IAP/MAP separation effect here simply reflect the increased temporal proximity of the firing of cells that fire during a run to a reward or at the reward location relative to mixed pairs across both populations (i.e. nothing to do with behavioral state, which just happens to be confounded with temporal proximity)? To answer this question requires plots of probability of co-reactivation versus the temporal distance between firing during behavior (e.g. assessed by cross correlogram peaks) for all three types of pairs (MAP-MAP, MAP-IAP, IAP-IAP).

3) Same as Q1 but for the separation in PFC firing patterns (is it just a spatial proximity effect?)

4) Same as Q2 but for the separation in PFC firing patterns (is it just a temporal proximity effect?)

5) Are IAPs different to place cells that happen to fire at a specific reward location?

6) Do IAP reactivation events control for the firing of IAPs due to being immobile at the reward location? And does IAP definition control for firing that happens during SWRs?

*Reviewer #2:*

The manuscript by Yu et al. describes a study investigating reactivation patterns of place cells associated with immobility (IAP, first described by Kay et al. (2016)) and those associated with mobility (MAP). The authors find the two cell types are active in distinct SWRs and the degree of co-activity is less than one would expect by chance. Further, in a different experiment the authors find PFC activity is coordinated with CA1 activity during reactivations, such that distinct sets of PFC cells are recruited by IAP and MAP SWRs. Finally, the authors explain the separation of mobility and immobility representations can be accounted for by a Hebbian mechanism. This work represents an advance to our nascent understanding of place cells associated with immobility. However, I have some reservations about the methods used to highlight differences between IAP and MAPs and the overall novelty of the findings.

1) The authors define IAPs and MAPs based on the animals' location – cells active at the reward wells are IAPs and cells active elsewhere are MAPs. They seek to address whether a transition from a mobility representation to an immobility representation occurs smoothly or abruptly, using pair-wise cross-correlations, and find evidence for a smooth transition. My only issue with this result is that it has to be largely influenced by the parameters used. Namely, the IAPs, although they fire mostly during immobility, they also fire during movement; the opposite is true for MAPs. Thus, the authors are inclined to find a smooth transition.

2) Differences in SWR power and duration could be explained by number of cells/spikes active in IAP and MAP SWRs. Have the authors tried to control for this? Given there are much fewer IAPs than MAPs I assume there are fewer cells/spikes active in IAP SWRs.

3) The modulation of PFC neurons by SWRs is basically a replication of the Jadhav et al. (2016) study showing coordinated reactivation between PFC and CA1. Showing PFC neurons are differentially recruited by IAP and MAP SWRs could just be explained by co-activity during tasks and that reactivations are recruited by CA1 (as indicated by Jadhav et al).

4) The section on the Hebbian mechanism which could account for the described representational segregation can go into the Discussion, or be combined with the previous section and Figure 7 be included as a supplement. It doesn't show a result, but rather is a mechanistic description of the effect.

5) Is there any temporal structure to IAP and MAP SWRs? i.e. to IAP SWRs immediately precede/follow MAP SWRs? This would support the claim that IAPs are involved in replay of experiences. Presumably, MAP SWRs reactivate trajectories on the tracks, and each trajectory is flanked by a period of immobility. Thus, one might expect to observe this structure in the SWRs.

*Reviewer #3:*

This manuscript shows that CA1 cells that fire preferentially in a place-related manner during immobility (IAP) tend to fire together during sharp wave ripples (SWR) as well that occur during exploration. Moreover, in some SWRs place cells fire preferentially together while in other SWRs IAP cells are co-active together. In the prefrontal cortex some cells preferentially increase their firing with place cell firing-related SWRs whereas, other prefrontal cell may fire with IAP-associated SWRs. The work is important: it examined the reactivation properties of IAPs and its link to prefrontal activity. I have one concern that I am sure the authors can address.

1) It has not been tested whether IAP-associated SWRs exist in significant numbers that express a location other than the current location of the animal. It was shown that IAPs encoding similar locations co-fire together stronger during SWRs than those encoding different locations. It was also shown that IAPs that encode a location different from the current location of the animal exhibit correlated firing with IAP-SWRs overall. These two findings indicate but does not directly show that IAP-associated SWRs exist that on the assembly level encode a location other than the current location of the animal. This would be important to show otherwise one may not be able to refer to these events as reactivation. SWRs may just reflect the high frequency leakage of intense spiking in some periods when IAPs encoding the current location transiently increase their firing. Such processes may just reflect sensory-driven encoding of the location of the animal with transient synchronization, not reactivation.

[Editors' note: further revisions were requested prior to acceptance, as described below.]

Thank you for resubmitting your work entitled "Distinct hippocampal-cortical memory representations for experiences associated with movement versus immobility" for further consideration at *eLife*. Your revised article has been favorably evaluated by Eve Marder (Senior Editor), a Reviewing Editor, and two reviewers.

The manuscript has been substantially improved and all major issues addressed. There is a single outstanding issue from reviewer 2 which remains to be addressed:

The authors have shown IAP SWRs have fewer cells and spikes active in them than MAP SWRs. They also report IAP SWRs are lower in amplitude and shorter in duration. Could the authors control for these difference, e.g. with a down-sampling procedure, to show the amplitude/duration is not simply a product of number of cells/spikes differences. As it stands, it is not clear whether differences in the two kinds of SWRs represents a physiological difference between IAP and MAP SWRs or simply a difference in the number of spikes for the two SWR types.

---

## [Author Response]

*Reviewer #1:*

*[…] As I read the paper several basic questions came to mind that needed to be resolved before the results could be interpreted or evaluated properly (particularly Qs 1 and 2 below). These may be answered within the paper, but if so not in such an obvious way that I could tell.*

We thank reviewer #1 for raising important issues to improve our manuscript. We have edited the text to address the controls proposed.

*1) Given that reactivation of firing during SWRs is most commonly thought of in terms of 'replay' of experience, and such 'replay events' correspond to experiential trajectories, one would expect the probability of reactivation of a pair of cells to depend on the spatial proximity of their firing fields. Does the IAP/MAP separation effect here simply reflect the increased spatial proximity of the firing fields of cells that fire during a run to a reward or at the reward location relative to mixed pairs across both populations (i.e. nothing to do with behavioral state, which just happens to be confounded with spatial proximity)? To answer this question requires plots of probability of co-reactivation versus the spatial distance between firing fields for all three types of pairs (MAP-MAP, MAP-IAP, IAP-IAP).*

We agree with reviewer #1 that place field spatial proximity is related to coactivity between place cells during SWRs. We therefore added scatter plots of distance between place field peaks for each cell pair and their SWR coactivity as well as binned quantifications of the values for small distance separations for each group (Figure 4—figure supplement 2). These results are consistent with previous findings, where place cells with place fields further apart in space are less coactive during SWRs than those with nearby place fields.

Critically, as shown in the Figure 4—figure supplement 2, SWR coactivity for IAP-MAP pairs was significantly lower than for matched MAP-MAP pairs across a wide range of place field peak distances. We show the boxplots for place field centers <0.5m apart in panel C. These findings demonstrate that spatial proximity cannot fully account for the IAP/MAP SWR separation effect.

We also note that because our environment geometry allows travel on multiple paths between the same locations, place field peak distance could in principle be dissociated from the time between spikes of pairs of cells. Thus, temporal proximity, which reviewer #1 has also suggested, is likely to be more important when comparing SWR coactivity in our tasks.

*2) Given that reactivation of firing during SWRs is most commonly thought of in terms of 'replay' of experience, and such 'replay events' correspond to experiential trajectories, one would expect the probability of reactivation of a pair of cells to depend on the temporal proximity of their firing during experience. Does the IAP/MAP separation effect here simply reflect the increased temporal proximity of the firing of cells that fire during a run to a reward or at the reward location relative to mixed pairs across both populations (i.e. nothing to do with behavioral state, which just happens to be confounded with temporal proximity)? To answer this question requires plots of probability of co-reactivation versus the temporal distance between firing during behavior (e.g. assessed by cross correlogram peaks) for all three types of pairs (MAP-MAP, MAP-IAP, IAP-IAP).*

We entirely agree that temporal proximity of spiking is a critical determinant of SWR coactivity, and indeed the cartoon model we present in Figure 7 makes that point. Proximity alone cannot explain our findings, however. To make that clear, we added scatter plots to show the offset to the peak of the cross-correlation within a 20s window and the SWR coactivity for each cell pair (Figure 4—figure supplement 1 and Figure 4). As expected, and consistent with a role for temporal proximity, we saw cells that are active closer in time are more likely to be coactive during SWRs.

At the same time, when we matched pairs for temporal proximity, we found that the SWR coactivity for IAP-MAP pairs was lower compared with MAP-MAP pairs across a range of offset times. Here we note that this difference was significant for offsets > 1 second but was not for offsets of less than 1 second. This small group of MAP-IAP pairs with small offsets in cross-correlation peaks likely accounts for associations between MAP-IAPs that contribute to the small number of joint MAP-IAP SWRs. This is consistent with our proposed model where the amount of temporal overlap in spiking between two cells relative to all spikes from each cell, rather than the absolute temporal offset between all spikes, affects their association. We should emphasize, however, that the highly significant differences for larger offsets demonstrates that temporal offset between spiking alone cannot fully account for the IAP/MAP separation effect. We now clarify this point in the last paragraph of the subsection “Separate reactivation of immobility-associated spatial representations”.

*3) Same as Q1 but for the separation in PFC firing patterns (is it just a spatial proximity effect?)*

*4) Same as Q2 but for the separation in PFC firing patterns (is it just a temporal proximity effect?)*

Please see Figure 6—figure supplement 2.

*5) Are IAPs different to place cells that happen to fire at a specific reward location?*

IAPs are similar to place cells that happen to fire at specific reward locations. In our case we cannot determine whether IAPs encode only location or location of reward, but we note that previous work from our laboratory (Kay et al. 2016) demonstrated that cells that showed place specific activity during immobility were also active at locations where no reward was delivered. Thus, we suggest IAPs are place cells that participate in representations for specific locations where the animal remains immobile, which in our task corresponds to specific locations where reward is delivered. To explain this in detail we included a section in the Discussion focusing on similarities and differences between IAPs and previously described goal or reward related activity in the hippocampus.

“The IAPs we describe here share similar properties to N-wave coupled cells that participate in immobility spatial coding (Kay et al., 2016) or have spatially selective activity during immobile waiting periods (McKenzie et al., 2013). […] This contrasts with previously described goal-related place cells that have dominant place fields away from the goal location (Hok et al., 2007).”

*6) Do IAP reactivation events control for the firing of IAPs due to being immobile at the reward location? And does IAP definition control for firing that happens during SWRs?*

This is an important question. We first note that the cross-correlation of IAP spiking with SWR event occurring within the place field of the IAP (in field SWRs), showed a significant increase in SWR aligned firing compared with baseline control, where the times of spiking were randomly shifted by ± 0-500ms (now shown in Figure 3—figure supplement 2). This means that the spiking during SWRs cannot be accounted for simply by the in-field firing of the IAP cell.

The question, then, is how to “control” for the place field activity. For example, as the reviewer is perhaps suggesting, it is possible that the firing during the SWR is best understood as a combination of place field input and SWR-related reactivation. Alternatively, it could be that IAP activity during these SWRs represents a switch to a different network state where local place field activity has at most a minor effect on spiking. There are a number of reasons to think that the second of these two possibilities (that SWRs occur during a different network state) is the most likely. First, as shown in Figure 3—figure supplement 2, there are decreases in firing rate relative to the place field baseline at the beginning and end of the SWR period, suggesting that a simple addition of place field activity and SWR excitation cannot explain the results. Second, we previously described CA2 immobility spatial coding cells that become transiently inhibited during SWRs (Kay et al., 2016), again suggesting that SWRs are momentary interruptions in the ongoing network state.

Those arguments aside, our SWR coactivity measure takes into account any baseline spiking during SWRs since the coactivity is normalized to both cells’ expected probability of spiking during SWRs, which is determined by repeated permutation of SWR identity to estimate “chance” coactivity (please see subsection “SWR coactivity” in the Methods).

While we cannot rule out a local spatial input as a contributor to SWR spiking, our results suggest this potential input cannot account for the magnitude of firing we observed and our analytical methods accounts for this baseline level of activation in any case.

To address the converse issue, we defined IAPs based on firing rates during immobility but only for times outside of SWRs (please see subsection “Cell selection” in Methods), thus spiking during SWRs does not contribute to the definition of IAPs.

*Reviewer #2:*

*[…] This work represents an advance to our nascent understanding of place cells associated with immobility. However, I have some reservations about the methods used to highlight differences between IAP and MAPs and the overall novelty of the findings.*

*1) The authors define IAPs and MAPs based on the animals' location – cells active at the reward wells are IAPs and cells active elsewhere are MAPs. They seek to address whether a transition from a mobility representation to an immobility representation occurs smoothly or abruptly, using pair-wise cross-correlations, and find evidence for a smooth transition. My only issue with this result is that it has to be largely influenced by the parameters used. Namely, the IAPs, although they fire mostly during immobility, they also fire during movement; the opposite is true for MAPs. Thus, the authors are inclined to find a smooth transition.*

First, we feel it is important to emphasize that in our previous description of cells with place specific activity during immobility (Kay et al., 2016), we focused on periods where animals had been immobile for >2 seconds. Thus, it was possible that these cells were only active only during sustained immobility, and that their activity did not, in fact, overlap in time with the activity of movement-associated place cells. Alternatively, it was conceivable that we would see a very abrupt transition given that spatial representations expressed in the hippocampus on the time scale of theta can switch dramatically between consecutive theta cycles (Jezek et al., 2011). We have now clarified this possibility in the text:

“However, it is unknown whether transitions between movement and immobility spatial representations are smooth or abrupt. […] In this case, MAP spatial representations would be replaced by IAP spatial representations and vice versa, resulting in no overlap between MAP and IAP spiking in time (Figure 2).”

We thus had to examine the activity of these neurons without an explicit cutoff that excluded transitional times. When we did so, we found that cells that were most active during immobility could also be active during periods immediately preceding and following bouts of movement, which did not have to be the case. While this may seem obvious in retrospect, the results could have been different and these analyses were necessary to determine the temporal structure of IAP and MAP activity during the transition periods.

*2) Differences in SWR power and duration could be explained by number of cells/spikes active in IAP and MAP SWRs. Have the authors tried to control for this? Given there are much fewer IAPs than MAPs I assume there are fewer cells/spikes active in IAP SWRs.*

We entirely agree with reviewer #2 that the number of spikes affects the length and amplitude of SWRs. Our aim was to highlight that IAP-SWRs are smaller in amplitude and shorter in duration than MAP-SWRs or IAP and MAP -SWRs. This was the unexpected finding and we want to emphasize that in past work smaller amplitude SWRs were typically discarded from analysis. To clarify this point we have now added Figure 4—figure supplement 4 showing the mean cell and spike number during SWRs.

*3) The modulation of PFC neurons by SWRs is basically a replication of the Jadhav et al. (2016) study showing coordinated reactivation between PFC and CA1. Showing PFC neurons are differentially recruited by IAP and MAP SWRs could just be explained by co-activity during tasks and that reactivations are recruited by CA1 (as indicated by Jadhav et al).*

Here we think that we did not explain the differences between these findings and our previous work sufficiently. Our aim was not to replicate the finding of Jadhav et al., but to use those findings, which indicate PFC and hippocampal activity are coordinated, to further validate the existence of specific reactivations of representations associated with immobility across both structures.

In particular, our findings provide three main advances from this previous work:

A) PFC cells show contrasting modulation directions associated with different reactivated hippocampal representations.

Jadhav et al. found a subset of cells showed excitation while others showed inhibition during SWRs. This suggested PFC cells show stereotyped changes, either excitation or inhibition, during SWRs. This was based the modulation to all SWRs, without classifying their content.

In the current study, we show that activity of single PFC cells is differentially modulated around IAP- and MAP-SWRs. The most striking and key finding is that modulation of some PFC cells distinguishes between hippocampal immobility and movement reactivations by showing contrasting modulation directions. This is consistent with our previous work, but it is an unexpected pattern of change during SWRs. This finding is also novel in the context of other published results in rodent PFC (Remondes and Wilson, 2015; Siapas and Wilson, 1998; Wang and Ikemoto, 2016; Wierzynski et al., 2009). In all of these studies, PFC modulation during SWRs was examined for all SWRs without knowledge of potentially different SWR representations (IAP- or MAP-SWRs).

B) SWR related changes in PFC distinguish between movement and immobility associated hippocampal reactivations.

Jadhav et al. only examined coordination with CA1 activity using times when the animal was moving (speed >5cm/s). The findings of that paper indicated that positive speed-correlated PFC cells, or movement associated representations, were excited during SWRs while negative speed-correlated PFC cells, or immobility associated representations, were inhibited.

Our findings go beyond Jadhav et al. to identify a unique subset of PFC cells that show excitation during IAP-SWRs but not MAP-SWRs. This indicates immobility associated representations in PFC can also be reactivated during SWRs. There was no way to identify these two categories from Jadhav et al. and thus no way to conclude whether or how movement and immobility associated PFC representations might be reactivated during SWRs.

Furthermore, transient excitation of PFC activity during SWRs that is associated with the reinstatement of representations is selective to one type of SWR. We found significantly fewer PFC cells that show excitation during both types of SWRs. This result demonstrates movement and immobility associated PFC representations are distinct, since very few PFC cells participate in the reinstate of both representations. Additionally, this reaffirms the distinction between movement and immobility SWRs that we found in the hippocampus and this distinction is also preserved in PFC.

C) The patterns of SWR related changes in PFC are consistent with their ongoing non-SWR activity and coordination with CA1.

Jadhav et al. characterized the relationship between PFC-CA1 coordination during SWRs and outside of SWRs but only with respect to periods of movement. This was done by limiting spiking data to times when the animal was moving >5cm/s or when hippocampal theta was prominent. This was carried out for both SWR inhibited and excited PFC cells. This filtering of the data precludes any direct statements about PFC-CA1 activity for immobility associated representations. Our current work extends this finding to immobility periods and also goes beyond Jadhav et al. in another important way: we identified a specific group of cells that are inhibited during both immobility and movement SWRs and found that these cells show a pattern of ongoing activity during the task that does not overlap strongly with either IAPs or MAPs in the hippocampus. While this is certainly broadly consistent with our previous demonstration of coordinated reactivation across structures, the separate reactivation of immobility and movement related representations and the concomitant suppression of this distinct type of activity cannot be derived directly from our previous findings.

*4) The section on the Hebbian mechanism which could account for the described representational segregation can go into the Discussion, or be combined with the previous section and Figure 7 be included as a supplement. It doesn't show a result, but rather is a mechanistic description of the effect.*

We agree and have moved this to the Discussion.

*5) Is there any temporal structure to IAP and MAP SWRs? i.e. to IAP SWRs immediately precede/follow MAP SWRs? This would support the claim that IAPs are involved in replay of experiences. Presumably, MAP SWRs reactivate trajectories on the tracks, and each trajectory is flanked by a period of immobility. Thus, one might expect to observe this structure in the SWRs.*

We calculated the cross-correlation between MAP-SWRs and IAP-SWRs but we do not find any strong relationship between the timing of MAP-SWRs and IAP-SWRs (now shown in Figure 4—figure supplement 6). However, in SWRs that contain both IAPs and MAPs, we found IAPs generally spike earlier than MAPs (Figure 4 and subsection “Distinct properties for SWRs involving movement and immobility representations”, last paragraph) while controlling for the duration of SWRs. This suggests although MAP-SWRs and IAP-SWRs are independent events, joint events are likely to reactivate trajectory representations originating from a location of immobility represented by IAPs.

*Reviewer #3:*

*[…] I have one concern that I am sure the authors can address.*

*1) It has not been tested whether IAP-associated SWRs exist in significant numbers that express a location other than the current location of the animal. It was shown that IAPs encoding similar locations co-fire together stronger during SWRs than those encoding different locations. It was also shown that IAPs that encode a location different from the current location of the animal exhibit correlated firing with IAP-SWRs overall. These two findings indicate but does not directly show that IAP-associated SWRs exist that on the assembly level encode a location other than the current location of the animal. This would be important to show otherwise one may not be able to refer to these events as reactivation. SWRs may just reflect the high frequency leakage of intense spiking in some periods when IAPs encoding the current location transiently increase their firing. Such processes may just reflect sensory-driven encoding of the location of the animal with transient synchronization, not reactivation.*

We thank reviewer #3 for this insightful suggestion. For IAP pairs that are active at the same well location, we calculated their SWR coactivity during SWRs away from the well location where the IAPs were most active (not in field SWRs). The coactivity of IAP pairs during SWRs away from the current location of the animal is not significantly different from all SWRs, both of which are significantly higher than IAP pairs active at different locations (now shown in Figure 3—figure supplement 3). This suggests IAP pairs encoding similar locations are coactive during non-local SWRs to reactivate non-local immobility location representations.

[Editors' note: further revisions were requested prior to acceptance, as described below.]

*There is a single outstanding issue from reviewer 2 which remains to be addressed:*

*The authors have shown IAP SWRs have fewer cells and spikes active in them than MAP SWRs. They also report IAP SWRs are lower in amplitude and shorter in duration. Could the authors control for these difference, e.g. with a down-sampling procedure, to show the amplitude/duration is not simply a product of number of cells/spikes differences. As it stands, it is not clear whether differences in the two kinds of SWRs represents a physiological difference between IAP and MAP SWRs or simply a difference in the number of spikes for the two SWR types.*

We have resampled MAP- and IAP-SWR populations as suggested. The durations and normalized power of MAP-SWRs still remain longer and higher, respectively, compared with IAP-SWRs. We have added the two controls as Figure 4—figure supplement 6-7 and edited the text to reflect these changes.